# Unsupervised Learning for Class Distribution Mismatch

**Pan Du**[1]  **Wangbo Zhao**[†2]  **Xinai Lu**[3]  **Nian Liu**[4]  **Zhikai Li**[5]  **Chaoyu Gong**[2]  **Suyun Zhao**[†1]  **Hong Chen**[1]
**Cuiping Li**[1]  **Kai Wang**[2]  **Yang You**[2]

## Abstract

Class distribution mismatch (CDM) refers to the discrepancy between class distributions in training data and target tasks. Previous methods address this by designing classifiers to categorize classes known during training, while grouping unknown or new classes into an "other" category. However, they focus on semi-supervised scenarios and heavily rely on labeled data, limiting their applicability and performance. To address this, we propose **U**nsupervised Learning for **C**lass **D**istribution **M**ismatch (UCDM), which constructs positive-negative pairs from unlabeled data for classifier training. Our approach randomly samples images and uses a diffusion model to add or erase semantic classes, synthesizing diverse training pairs. Additionally, we introduce a confidence-based labeling mechanism that iteratively assigns pseudo-labels to valuable real-world data and incorporates them into the training process. Extensive experiments on three datasets demonstrate UCDM's superiority over previous semi-supervised methods. Specifically, with a 60% mismatch proportion on Tiny-ImageNet dataset, our approach, without relying on labeled data, surpasses OpenMatch (with 40 labels per class) by 35.1%, 63.7%, and 72.5% in classifying known, unknown, and new classes.

## 1. Introduction

Class distribution mismatch (CDM) (Guo et al., 2020; Saito et al., 2021; Du et al., 2022; Li et al., 2023) has garnered

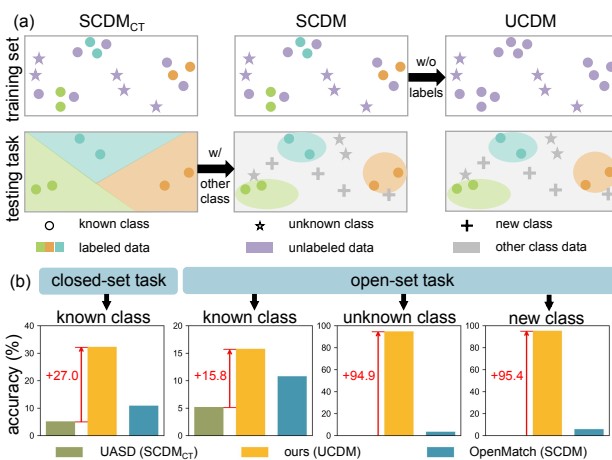

*Figure 1.* (a) Examples of SSL for closed-set task (SCDM$_{CT}$), open-set task (SCDM), and our proposed unsupervised learning for class distribution mismatch (UCDM), where no labels are used during training. (b) Accuracy of methods on closed-set and open-set tasks. In the closed-set task, samples are classified into known classes, while in the open-set task, they may be classified as unified "other" class, including both unknown and new categories.

increasing attention in recent years. It tracks the practical problem where the class distribution of the available training dataset fails to align with the requirements of the target task (Yang et al., 2022; Fan et al., 2023; Ma et al., 2023).

Previous researches on CDM primarily concentrate on semi-supervised learning (SSL), which requires access to both labeled and unlabeled data (Berthelot et al., 2019; Sohn et al., 2020). In this context, the categories present in the labeled data are referred to as *known classes*, while the unlabeled data contains not only the known classes but also additional *unknown classes* that are absent in the labeled data. As illustrated in Fig. 1 (a), based on differences in the target tasks, these SSL methods fall into two groups: *(i)* The first group focuses on a Closed-set Task (SCDM$_{CT}$), where the goal is to *classify instances solely among the known classes present in the labeled data* (Chen et al., 2020; Guo et al., 2020; Huang et al., 2021; Yang et al., 2022). Mainstream approaches typically tackle SCDM$_{CT}$ challenge by filtering out unknown categories from the unlabeled data, thus mitigating their negative influence. *(ii)* The second group extends SCDM$_{CT}$ to the Open-set Task (SCDM), where the

---

[†]Corresponding author [1]School of Information, Renmin University of China, and Engineering Research Center of Database and Business Intelligence, MOE, China [2]National University of Singapore [3]School of Agricultural Economics and Rural Development, Renmin University of China [4]Independent Researcher [5]Institute of Automation, Chinese Academy of Sciences. Correspondence to: Wangbo Zhao <wangbo.zhao96@gmail.com>, Suyun Zhao <zhaosuyun@ruc.edu.cn>, Pan Du <du_pan@163.com>.

*Proceedings of the 42$^{nd}$ International Conference on Machine Learning*, Vancouver, Canada. PMLR 267, 2025. Copyright 2025 by the author(s).

objective extends beyond classifying known classes to *also identifying unknown classes and any new categories* that are absent from the labeled data as a unified "other" class during testing (Saito et al., 2021; Li et al., 2023). Methods in this category often employ one-vs-all classifiers to distinguish whether instances belong to known classes.

However, the requirement for labeled data constrains both the application and performance of these approaches. First, their heavy reliance on labeled data renders them impractical in scenarios where ground truth labels are unavailable. This dependency necessitates significant human and financial costs and may even require domain-specific knowledge (Gidaris et al., 2018; Zhang et al., 2023a). More importantly, training with limited labeled data restricts their performance to capture key features of known classes when extending $SCDM_{CT}$ to open-set tasks, as SCDM methods (Saito et al., 2021; Li et al., 2023). This is because they primarily train one-vs-all classifiers by treating labeled instances from a specific known class as positives and all other labeled instances as negatives. As a result, the model struggles to distinguish samples from unknown categories outside the data manifold, leading to unstable performance. As illustrated in Fig. 1 (b), on the open-set task of the Tiny-ImageNet (Deng et al., 2009) dataset, OpenMatch (Saito et al., 2021), a representative SCDM method, effectively classifies known-classes instances, while its performance for consolidating unknown and new classes into a unified "other" class is notably subpar. Hence, developing methods for open-set tasks without relying on ground truth labels is imperative.

To alleviate this problem, we introduce **U**nsupervised Learning for **C**lass **D**istribution **M**ismatch (UCDM), which operates without ground truth labels in the training data and utilizes only a predefined set of class names from known classes. In this context, we aim to construct positive-negative pairs for training the classifier without any human annotation, adhering to the unsupervised learning setting (Goodfellow et al., 2016). First, we theoretically demonstrate that diffusion models (Ho et al., 2020) can erase semantic classes from images. Given an original image, this capability enables us to generate negative instances by removing the semantic class from the image, corresponding to the positive instances guided by class prompts. Subsequently, through a confidence-based labeling mechanism, valuable real images are paired with the generated images to incorporate them into the training process. This approach effectively mitigates the reliance on labeled data and provides a training framework to tackle the CDM problem.

Extensive experiments on diverse datasets, including CIFAR-10 (Krizhevsky et al., 2009), CIFAR-100 (Krizhevsky et al., 2009), and Tiny-ImageNet (Deng et al., 2009), demonstrate that our method achieves superior performance without relying on labeled data. Notably, on Tiny-ImageNet with 60% mismatch proportion as shown in Fig. 1 (b), our approach outperforms UASD (Chen et al., 2020), a $SCDM_{CT}$ method, by 27.0% on the closed-set task. For open-set tasks, our approach surpasses Open-Match (Saito et al., 2021), a classic SCDM method, by 5.0%, 91.4%, and 89.5% in the classification of known, unknown, and new classes, respectively. These results highlight the robustness of UCDM in both closed-set and open-set tasks, positioning it as a promising direction for advancing CDM.

## 2. Related Work

### 2.1. SSL under Class Distribution Mismatch

SSL methods for class distribution mismatch can be divided into two branches: one addressing closed-set tasks ($SCDM_{CT}$) and another tackling open-set tasks (SCDM).

$SCDM_{CT}$ methods train classifiers to classify known-class instances by filtering unknown-class samples from unlabeled data. Prediction consistency is exploited by $DS^3L$ (Guo et al., 2020), which identifies consistency loss discrepancies between augmented views, and UASD (Chen et al., 2020), which averages predictions from a temporally ensembled classifier. Confidence-based methods, such as CCSSL (Yang et al., 2022) and SSB (Fan et al., 2023), classify instances with maximum probabilities below a threshold as unknown. T2T (Huang et al., 2021) judge whether image embeddings align with pseudo-labels using a cross-modal matching model, while OSP (Wang et al., 2023) extend it by excluding unknown-class pixels from features.

SCDM methods classify known-class instances while grouping unknown or new-class instances into a unified "other" class. MTCF (Yu et al., 2020) trains a detector to distinguish known from other classes. A prototype-based approach (Ma et al., 2023) builds prototypes for unknown-class instances using distance functions. OpenMatch (Saito et al., 2021) and IOMatch (Li et al., 2023) use multi-binary classifiers trained in a one-vs-all manner, treating known-class instances as positives and others as negatives. Combining these outputs with a closed-set classifier generates a probability distribution over known and other classes.

However, the reliance on labeled data limits both the applicability and performance of these methods, while unsupervised learning settings remain unexplored.

### 2.2. Diffusion-Based Generation Methods

Image synthesis (Azizi et al., 2023; Dai et al., 2023; Tian et al., 2024) has gained significant attention due to the ability of diffusion models (Rombach et al., 2022; Ramesh et al., 2022; Saharia et al., 2022) to generate high-quality data.

A classic strategy in image synthesis involves enriching prompts (Dunlap et al., 2023; Sarıyıldız et al., 2023; Shipard

et al., 2023) to guide the diffusion model, thereby expanding datasets. Another approach modifies real image embeddings by injecting learnable perturbations, creating variants enriched with novel information (Zhang et al., 2023b).

Recently, diffusion models have been integrated with SSL. DPT (You et al., 2024) employs a text-to-image paradigm, establishing a cyclical process where the SSL classifier is retrained with generated samples, and the updated classifier produces pseudo-labels to further train the diffusion model. However, it assumes matching class distributions between training data and the target task. Similarly, DWD (Ban et al., 2024) enhances known-class classification by training a diffusion model with labeled and unlabeled data to transform irrelevant unlabeled samples into known classes.

Our approach differs fundamentally from these methods. First, it avoids retraining the diffusion model. Second, it generates positive and negative instances without human annotations, enabling classifier training in the UCDM setting.

# 3. Unsupervised Learning for CDM

Sec. 3.1 provides an overview of diffusion probabilistic models, followed by the UCDM problem definition in Sec. 3.2. Sec. 3.3 and Sec. 3.4 introduce the positive and negative instance generation pipelines using diffusion models, and Sec. 3.5 details classifier training using generated instances.

## 3.1. Preliminary

**Diffusion probabilistic models (DPMs)** (Ho et al., 2020; Rombach et al., 2022; Ramesh et al., 2022; Saharia et al., 2022; Zhao et al., 2024b; 2025), involve a forward diffusion process and a reverse denoising process. Given a sample $\boldsymbol{x}$, the forward process gradually adds Gaussian noise to $\boldsymbol{x}$ to produce $\boldsymbol{x}_t$ as $t$ increases from $0$ until $T$, which can be formulated as:

$$\boldsymbol{x}_t = \sqrt{\alpha_t}\boldsymbol{x}_0 + \sqrt{1-\alpha_t}\boldsymbol{\epsilon}, \quad \boldsymbol{\epsilon} \sim \mathcal{N}(0,1), \quad (1)$$

where $\alpha_t = \prod_{i=1}^{t}(1-\beta_i)$ and $\{\beta_i\}_{i=0}^{T}$ denotes a fixed or learned variance schedule. In the reverse process, noise is removed from $\boldsymbol{x}_t$ using a learned noise estimator $\epsilon_\theta(\boldsymbol{x}_t, t, \mathcal{C})$ conditioned on $\mathcal{C}$, yielding $\boldsymbol{x}_{t-1}$ as:

$$\boldsymbol{x}_{t-1} = \sqrt{\frac{\alpha_{t-1}}{\alpha_t}}\boldsymbol{x}_t - \sqrt{\alpha_{t-1}}\psi(\alpha_t, \alpha_{t-1}, \sigma_t)\hat{\epsilon}_\theta(\boldsymbol{x}_t, t, \mathcal{C}) + \sigma_t\epsilon_t, \quad (2)$$

where $\psi(\alpha_t, \alpha_{t-1}, \sigma_t)$ denotes the constant schedule that depends on the fixed parameters $\alpha_t$, $\alpha_{t-1}$, and $\sigma_t$, and $\hat{\epsilon}_\theta(\boldsymbol{x}_t, t, \mathcal{C}) = \epsilon_\theta(\boldsymbol{x}_t, t) + \gamma[\epsilon_\theta(\boldsymbol{x}_t, t, \mathcal{C}) - \epsilon_\theta(\boldsymbol{x}_t, t)]$. Here, $\epsilon_\theta(\boldsymbol{x}_t, t)$ represents the DPM without condition, and $\gamma$ and $\sigma_t$ control the strength of conditional guidance and random noise $\epsilon_t$, respectively (Ho et al., 2020; Ho & Salimans, 2022). If $\sigma_t = 0$ and $\gamma = 0$ for all $t$, yielding the Denoising Diffusion Implicit Model (DDIM) (Song et al., 2020a).

**DPM's relationship to score-based generative models** has been well established in (Song et al., 2020b; Kim et al., 2022; Luo, 2022), which can be formulated as:

$$\epsilon_\theta(\boldsymbol{x}_t, t) = -\sqrt{1-\bar{\alpha}_t}\nabla_{\boldsymbol{x}_t}\log p_\theta(\boldsymbol{x}_t), \quad (3)$$

$$\epsilon_\theta(\boldsymbol{x}_t, t, \mathcal{C}) = -\sqrt{1-\bar{\alpha}_t}\nabla_{\boldsymbol{x}_t}\log p_\theta(\boldsymbol{x}_t \mid \mathcal{C}), \quad (4)$$

where $\bar{\alpha}_t = \prod_{i=0}^{t}\alpha_i$ and $p_\theta(\cdot)$ denotes the data distribution parameterized by $\theta$. $\nabla_{\boldsymbol{x}_t}\log p_\theta(\boldsymbol{x}_t)$ and $\nabla_{\boldsymbol{x}_t}\log p_\theta(\boldsymbol{x}_t \mid \mathcal{C})$ represent the gradient of the log-likelihood with respect to $\boldsymbol{x}_t$ in the unconditional and conditional settings, respectively. These gradients indicate the direction in the data space that maximizes the corresponding likelihood.

## 3.2. Problem Definition of UCDM

**Overview of training and testing datasets.** The training dataset $\mathcal{D}$ consists of unlabeled samples, with ground truth labels drawn from the label sets $\mathcal{Y}_{\text{known}}$ and $\mathcal{Y}_{\text{unknown}}$. In the proposed unsupervised learning for class distribution mismatch (UCDM), only a predefined set of class names from $\mathcal{Y}_{\text{known}}$ is accessible, while ground truth labels for $\mathcal{D}$ remain unavailable. Here, $\mathcal{Y}_{\text{known}} = \{1, 2, \ldots, K\}$ represents the set of $K$ known classes, while $\mathcal{Y}_{\text{unknown}}$ denotes the set of unknown classes, with $\mathcal{Y}_{\text{known}} \cap \mathcal{Y}_{\text{unknown}} = \emptyset$. Images in the test dataset includes categories from the known classes $\mathcal{Y}_{\text{known}}$, unknown classes $\mathcal{Y}_{\text{unknown}}$, and new classes $\mathcal{Y}_{\text{new}}$, where $\mathcal{Y}_{\text{new}} \cap (\mathcal{Y}_{\text{known}} \cup \mathcal{Y}_{\text{unknown}}) = \emptyset$.

**Classifier architecture.** Our ultimate goal is to accurately assign instances from known classes to $K$ distinct categories and group instances from both unknown and new classes into a unified "other" class. To achieve this, our target classifier comprises three components: *(i)* a shared feature encoder; *(ii)* a fully connected layer with a shape of $2K$, serving as an open-set classifier comprising $K$ binary classifiers. The $j$-th binary classifier predicts the probability that an instance belongs to known class $j$, denoted as $p(j \mid \boldsymbol{x})$; *(iii)* a fully connected layer with a shape of $K$, serving as a closed-set classifier, producing the probability $\hat{p}(j \mid \boldsymbol{x})$ for $K$-way classification in the closed-set task.

However, training each binary classifier requires both positive and negative instances, and the closed-set classifier requires positive instances - both of which are not available in our unsupervised scenario. To address this challenge, we propose a diffusion-driven instance generation method that effectively creates sufficient positive-negative pairs based on seed samples randomly drawn from the training set.

## 3.3. Positive Instance Generation

We consider that effective positive instances for training should meet the following criteria:

*(i)* The generated images should avoid domain shifts, remain-

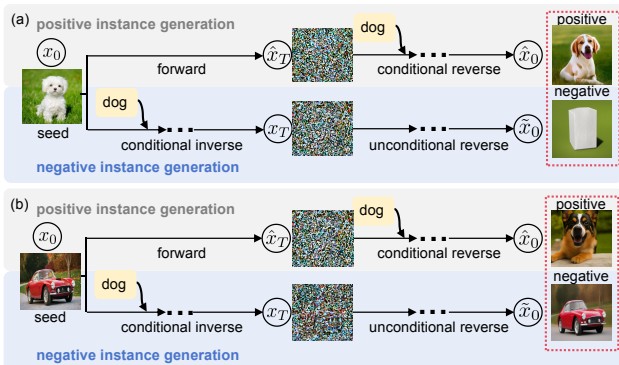

*Figure 2.* Pipelines for instance generation. (a) and (b) show that the semantic class in the prompt can be synthesized in the positive instance pipeline or erased in the negative instance pipeline for a given seed sample. If the seed sample lacks the specified semantic class, the generated image resembles the original image.

ing consistent with the characteristics of natural images.

*(ii)* The diversity of generated images should be sufficient within each category.

*(iii)* They must belong to one of the $K$ known classes with clearly identified categories.

To this end, we propose a diffusion-driven positive instance generation pipeline based on a text-to-image DPM model.

Specifically, to address property *(i)* and generate images as realistic training data, we randomly drive a sample from the training set and progressively add noise using Eq. (1) in the forward process to obtain a noise vector rather than a random one. This ensures that the noise vector forms a Gaussian centered around the seed sample, thereby preserving the information of the seed sample (Luo, 2022; He et al., 2022). To achieve property *(ii)*, we set $\sigma_t$ to 1 during the reverse process, introducing random noise in each step to enrich the diversity of the generated instances (Ho et al., 2020; Song et al., 2020a). Lastly, property *(iii)* is achieved through conditional generation guided by class prompt. Given a known class $y \in \mathcal{Y}_{\text{known}}$, we construct the prompt $\mathcal{C}_y$ as "A photo of a [CLASS]." where [CLASS] corresponds to the name of the class $y$. Then the conditional reverse process for positive instances is formulated as Eq. (2). As illustrated in Fig. 2, this ensures that the generated instances exhibit the target semantics without introducing domain shift.

### 3.4. Negative Instance Generation

To enhance the model's ability to capture key features of known classes and push samples from unknown classes outside the data manifold, the positive-negative pairs should provide effective contrast (Tack et al., 2020). Thus, negative instances should satisfy the following properties:

*(i)* They should belong to distinct semantic classes, differing

significantly from their corresponding positive instances.

*(ii)* They should share similar visual traits—such as structure and color—with their corresponding positives.

**Erasing semantic class via conditional inversion.** To achieve property *(i)*, we aim to erase the semantic class $y$, i.e., the class of positive instance, from the seed sample. Unconditional DDIM inversion (Song et al., 2020a; Kim et al., 2022) adds noise $\epsilon_\theta(\boldsymbol{x}_t, t)$ predicted by an unconditional DPM to the real image, mapping it to a latent vector from which the image can be reconstructed via unconditional reverse process. Hence, by modifying this inversion process to erase class-specific semantics, we satisfy property *(i)*.

The essence of erasing a semantic class lies in minimizing the likelihood of instance belonging to the positive instance's class $y$. Inspired by Eq. (4), we propose conditional DDIM inversion, which employs a conditional DPM (Ho & Salimans, 2022) to map a real image to a noise vector instead of a random one. As demonstrated in Theorem 3.1, this process approximately moves $\boldsymbol{x}_0$ in the negative gradient direction of $\sum_{i=0}^{t-1} [\nabla_{\boldsymbol{x}_i} \log p_\theta(\boldsymbol{x}_i)^{s_i} + \nabla_{\boldsymbol{x}_i} \log p_\theta(y \mid \boldsymbol{x}_i)^{s_i}]$, with step sizes regulated by the noise schedule $s_i$. Notably, $-\nabla_{\boldsymbol{x}_i} \log p_\theta(y \mid \boldsymbol{x}_i)$ represents the data-space gradient that reduces the likelihood of $\boldsymbol{x}_i$ belonging to $y$, thereby driving the noise vector to progressively diverge from the semantics of $y$. The detailed proof can be found in Appendix A.5.

**Theorem 3.1.** *(Conditional DDIM (Song et al., 2020a) inversion: progressive movement of the noise vector away from semantic class): Let $\boldsymbol{x}_t$ denote the noise vector at time step $t$ in the conditional inversion, and let $\mathcal{C}_y$ be the prompt of class $y$. Define $\delta_t = \epsilon_\theta(\boldsymbol{x}_t, t, \mathcal{C}_y) - \epsilon_\theta(\boldsymbol{x}_{t-1}, t, \mathcal{C}_y)$, where $\epsilon_\theta$ is a conditional DPM. When inverting the real image $\boldsymbol{x}_0$ to a noise vector $\boldsymbol{x}_t$ via conditional DDIM, i.e., setting $\sigma_t = 0$ and $\gamma = 1$ in Eq. (2), we obtain:*

$$\mathbf{x}_t = \sqrt{\alpha_t}\mathbf{x}_0 - \sum_{i=0}^{t-1} [\nabla_{\mathbf{x}_i} \log p_\theta(\mathbf{x}_i)^{s_i} + \nabla_{\mathbf{x}_i} \log p_\theta(y \mid \mathbf{x}_i)^{s_i}]$$
$$+ \sum_{i=0}^{t-1} \frac{s_i}{1 - \sqrt{\bar{\alpha}_{i+1}}} \delta_{i+1},$$

(5)

*where $s_i = \sqrt{\alpha_t(1 - \bar{\alpha}_{i+1})}\psi(\alpha_{i+1}, \alpha_i, 0)$ controls the magnitude of each gradient step based on the noise schedule.*

Hence, we adopt the conditional DDIM inversion process to erase the given semantic class, where the formula for $\boldsymbol{x}_t$ is derived from Theorem 3.1 and Eq. (2), as in Eq. (6). Since $\delta_t$ cannot be directly computed and is empirically shown to be negligible (Song et al., 2020a; Wallace et al., 2023), as further supported in the Appendix A.4, we approximate $\epsilon_\theta(\boldsymbol{x}_t, t, \mathcal{C}_y)$ by $\epsilon_\theta(\boldsymbol{x}_{t-1}, t, \mathcal{C}_y)$. Smaller values of $\delta_i$ indicate that $\mathbf{x}_t$ more closely follows the idealized trajectory

defined by the deterministic components.

$$\boldsymbol{x}_t = \sqrt{\frac{\alpha_t}{\alpha_{t-1}}}\boldsymbol{x}_{t-1} + \sqrt{\alpha_t}\psi(\alpha_t, \alpha_{t-1}, 0)\epsilon_\theta(\boldsymbol{x}_{t-1}, t, \mathcal{C}_y). \quad (6)$$

**Preserving visual characteristics in the unconditional reverse process.** Furthermore, the term $-\nabla_{\boldsymbol{x}_i}\log p(\boldsymbol{x}_i)^{s_i}$ in Theorem 3.1 drives $\boldsymbol{x}_i$ toward regions of lower log-likelihood probability density, causing a distribution shift from $\boldsymbol{x}_0$. This deviation disrupts the preservation of visual characteristics, in contrast to the property *(ii)*.

Inspired by Eq. (3), we reverse $\boldsymbol{x}_t$ using an unconditional DPM. As shown in Theorem 3.2, the resulting image $\tilde{\boldsymbol{x}}_0$ effectively preserves the visual features of $\boldsymbol{x}_0$ while approximately removing only the class-specific semantics, since $\tilde{\delta}_t$ and $\delta_t$ are negligible (Song et al., 2020a; Wallace et al., 2023), as detailed in Appendix A.4 and Appendix A.7.

**Theorem 3.2.** *(Unconditional DDIM reverse: progressive recovery of visual characteristics): Let $\tilde{\boldsymbol{x}}_0$ denote the image generated by the unconditional DDIM reverse process starting from the conditional inversion noise vector $\boldsymbol{x}_t$. Under the assumptions of Theorem 3.1, let $\tilde{\delta}_t = \epsilon_\theta(\tilde{\boldsymbol{x}}_t, t) - \epsilon_\theta(\tilde{\boldsymbol{x}}_{t-1}, t)$. Reversing $\boldsymbol{x}_t$ to $\tilde{\boldsymbol{x}}_0$ via unconditional DDIM with $\sigma_t = 0$ and $\gamma = 0$ in Eq. (2), we have:*

$$\tilde{\mathbf{x}}_0 = \mathbf{x}_0 - \frac{1}{\sqrt{\alpha_t}}\sum_{i=0}^{t-1}\nabla_{\mathbf{x}_i}\log p_\theta(y|\mathbf{x}_i)^{s_i}$$
$$+ \sum_{i=1}^{t-1}\sum_{j=i}^{t-1}\frac{s_i}{\sqrt{\alpha_t(1-\bar{\alpha}_{j+1})}}\left[\tilde{\delta}_{j+1} - \delta_{j+1}\right]. \quad (7)$$

Hence, supported by Theorem 3.2, we utilize unconditional DDIM to reverse the conditional noise vector $\boldsymbol{x}_t$ into a new image $\tilde{\boldsymbol{x}}_0$. In this process, $\tilde{\boldsymbol{x}}_{t-1}$ is formulated as Eq. (8), derived from Eq. (2) with $\gamma = 0$. To mitigate potential image degradation caused by semantic removal and preserve visual fidelity, we introduce random noise with $\sigma_t = 0.2$.

$$\tilde{\boldsymbol{x}}_{t-1} = \sqrt{\frac{\alpha_{t-1}}{\alpha_t}}\tilde{\boldsymbol{x}}_t - \sqrt{\alpha_{t-1}}\psi(\alpha_t, \alpha_{t-1}, \sigma_t)\epsilon_\theta(\tilde{\boldsymbol{x}}_t, t) + \sigma_t\epsilon_t. \quad (8)$$

The generation of negative instances is illustrated in Fig. 2, with the diffusion-driven process detailed in Algorithm1 of Appendix A.3. Theorems 3.1 and 3.2 confirm the reliability of both positive and negative instances, further supported by results in Sec. 4.3 and visualizations in Appendix B.13.

Consequently, with a randomly selected seed sample from the training set and a prompt for a known class $y$, we can generate a positive instance labeled $y$ and a negative instance not belonging to $y$, following Sec. 3.3 and Sec. 3.4, respectively. This enables the construction of positive ($\mathcal{D}_P$) and negative ($\mathcal{D}_N$) instance sets for subsequent training.

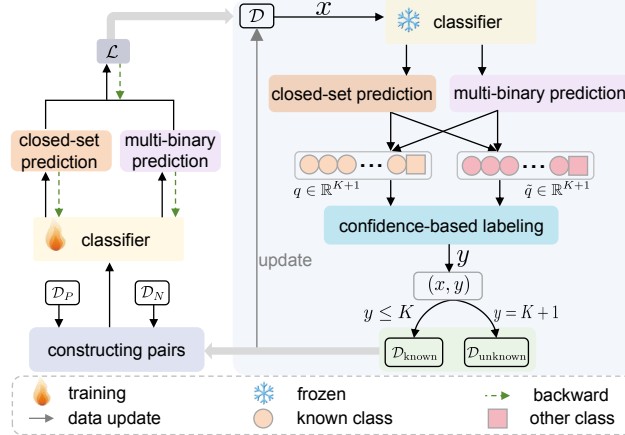

*Figure 3.* The framework for training an unsupervised classifier based on generated positive and negative instances.

## 3.5. Unsupervised Classifier Training

To differentiate known classes from unknown and new classes, we train the open-set classifier using both positive and negative instance sets. Specifically, we employ the loss function $\mathcal{L}_{\text{open}}^{(\mathcal{D}_P, \mathcal{D}_N)}$ to maximize the probability of positive instances being assigned to their respective classes, i.e., $p(y|\boldsymbol{x})$, while minimizing the probability of their corresponding negative counterparts.

To ensure the categorization of know classes, we constrain the open-set classifier with the loss function $\mathcal{L}_{\text{open}}^{\mathcal{D}_P}$ to assign the maximum probability to the corresponding class for each sample in the positive instance set $\mathcal{D}_P$. In addition, we also train a closed-set classifier $\mathcal{L}_{\text{closed}}^{\mathcal{D}_P}$ on the positive instance set $\mathcal{D}_P$ with the loss function to tackle the closed-set task.

Thus, the loss function for the generated data is defined as:

$$\mathcal{L}_{\text{generated}}^{(\mathcal{D}_P, \mathcal{D}_N)} = \lambda_1\mathcal{L}_{\text{open}}^{(\mathcal{D}_P, \mathcal{D}_N)} + \lambda_2\left[\mathcal{L}_{\text{open}}^{\mathcal{D}_P} + \mathcal{L}_{\text{closed}}^{\mathcal{D}_P}\right], \quad (9)$$

where $\lambda_1$ and $\lambda_2$ control the trade-off for each objective. For further details on each loss function, refer to Appendix A.1.

**Confidence-based labeling.** To leverage real images effectively, we propose a labeling mechanism that combines other-probability-driven and known-probability-driven confidences. Instances with high confidences are selected, assigned pseudo-labels, and incorporated into the training.

From the other-probability-driven perspective, an instance not belonging to any known class is assigned to the unified "other" class. The probability of this is computed by integrating the predictions of $K$ binary classifiers: $p(y \in \mathcal{Y}_{\text{other}} \mid \boldsymbol{x}) = \prod_{j=1}^{K}[1 - p(j \mid \boldsymbol{x})]$, where $p(j \mid \boldsymbol{x})$ is the probability that the $j$-th classifier predicts the sample as class $j$. Accordingly, the probability of belonging to a known class is $p(y \in \mathcal{Y}_{\text{known}} \mid \boldsymbol{x}) = 1 - p(y \in \mathcal{Y}_{\text{other}} \mid \boldsymbol{x})$.

The probability $\hat{p}(j \mid \boldsymbol{x})$ represents the likelihood of an

*Table 1.* The average accuracy of methods on the closed-set task across CIFAR-10, CIFAR-100, and Tiny-ImageNet datasets, with mismatch proportions ranging from 20% to 75%. The best and second-best results are highlighted in **bold** and underlined, respectively.

| method | CIFAR-10 | | | | CIFAR-100 | | | | Tiny-ImageNet | | | |
|---|---|---|---|---|---|---|---|---|---|---|---|---|
| | 20% | 40% | 60% | 75% | 20% | 40% | 60% | 75% | 20% | 40% | 60% | 75% |
| DS³L | 65.6 | 67.3 | 66.6 | 68.3 | 23.9 | 22.7 | 23.4 | 24.4 | 24.5 | 25.7 | 26.3 | 25.7 |
| UASD | 82.2 | 78.2 | 79.3 | 68.8 | 26.2 | 24.4 | 22.8 | 20.4 | 5.4 | 5.6 | 5.3 | 7.5 |
| CCSSL | **97.9** | **96.0** | **95.7** | 94.3 | 48.9 | 47.9 | 45.6 | 46.0 | 26.7 | 23.4 | 25.8 | 24.4 |
| T2T | - | - | - | - | 54.7 | 53.9 | 50.6 | 48.7 | **40.5** | **41.0** | **41.7** | **38.0** |
| MCTF | 62.0 | 64.7 | 61.2 | 71.8 | **56.3** | **55.6** | **56.6** | **56.6** | 29.1 | 29.4 | 23.1 | 26.2 |
| IOMatch | 96.6 | 92.9 | 89.8 | 86.1 | 29.4 | 30.3 | 31.1 | 32.4 | 31.4 | 32.9 | 32.8 | 32.8 |
| OpenMatch | 92.8 | 91.0 | 68.5 | 73.4 | 17.3 | 10.8 | 10.3 | 6.1 | 14.0 | 10.3 | 10.9 | 12.2 |
| Ours | 95.2 | 93.5 | 95.6 | **96.7** | 53.7 | 49.3 | 50.9 | 49.9 | 36.9 | 32.4 | 32.3 | 35.4 |

*T2T is excluded from CIFAR-10 as it is not applicable to binary classification.

instance belonging to the $j$-th known class in the closed-set task. Thus, in the open-set task, the probability for the $j$-th known class is $p(y \in \mathcal{Y}_{\text{known}} \mid \boldsymbol{x}) \cdot \hat{p}(j \mid \boldsymbol{x})$, leading to the $K+1$-way distribution $\boldsymbol{q} \in \mathbb{R}^{K+1}$ for an instance, formulated as:

$$q_j = \begin{cases} p(y \in \mathcal{Y}_{\text{known}} \mid \boldsymbol{x}) \cdot \hat{p}(j \mid \boldsymbol{x}), & \text{if } 1 \le j \le K, \\ p(y \in \mathcal{Y}_{\text{other}} \mid \boldsymbol{x}), & \text{if } j = K+1. \end{cases} \quad (10)$$

From the known-probability-driven perspective (Li et al., 2023), both $\hat{p}(j \mid \boldsymbol{x})$ and $p(j \mid \boldsymbol{x})$ estimate the likelihood that an instance belongs to the $j$-th known class. The probability of an instance belonging to the $j$-th known class is $\hat{p}(j \mid \boldsymbol{x}) \times p(j \mid \boldsymbol{x})$, while the probability of it belonging to the "other" class is $1 - \sum_{j=1}^{K} \hat{p}(j \mid \boldsymbol{x}) \times p(j \mid \boldsymbol{x})$. Thus, the class probability distribution $\tilde{\boldsymbol{q}} \in \mathbb{R}^{K+1}$ in open-set task is:

$$\tilde{q}_j = \begin{cases} \hat{p}(j \mid \boldsymbol{x}) \times p(j \mid \boldsymbol{x}), & 1 \le j \le K, \\ 1 - \sum_{j=1}^{K} \hat{p}(j \mid \boldsymbol{x}) \times p(j \mid \boldsymbol{x}), & j = K+1. \end{cases} \quad (11)$$

If the top-confidence class in both $\boldsymbol{q}$ and $\tilde{\boldsymbol{q}}$ is the $j$-th class and their scores exceed a threshold $\delta$, a pseudo-label $j$ is assigned. The labeled sample is then added to $\mathcal{D}_{\text{known}}$ or $\mathcal{D}_{\text{unknown}}$, depending on whether $j$ is a known or unknown class, and removed from the original training set $\mathcal{D}$.

Meanwhile, for the known-class set $\mathcal{D}_{\text{known}}$, negative instances are selected from $\mathcal{D}_N$ to form $\mathcal{D}'_N$, while positive instances are selected from $\mathcal{D}_P$ for the unknown-class set $\mathcal{D}_{\text{unknown}}$, forming $\mathcal{D}'_P$. These selected instances are then removed from $\mathcal{D}_P$ and $\mathcal{D}_N$. The total loss is Eq. (12).

$$\mathcal{L} = \mathcal{L}_{\text{generated}}^{(\mathcal{D}_P, \mathcal{D}_N)} + \mathcal{L}_{\text{generated}}^{(\mathcal{D}_{\text{known}}, \mathcal{D}'_N)} + \mathcal{L}_{\text{generated}}^{(\mathcal{D}'_P, \mathcal{D}_{\text{unknown}})}. \quad (12)$$

The classifier training pipeline is shown in Fig. 3, with the Algorithm 2 in Appendix A.3; see Appendix A.2 for pair construction details.

# 4. Experiments

## 4.1. Experimental Setups

**Datasets.** Following previous works (Chen et al., 2020; Li et al., 2023), we employ three benchmark datasets, including CIFAR-10 (Krizhevsky et al., 2009), CIFAR-100 (Krizhevsky et al., 2009), and Tiny-ImageNet (Deng et al., 2009). More details please refer to Appendix C.1.

**Settings.** *(i)* We vary the mismatch proportion—i.e., the percentage of unknown-class instances in training data—across 0%, 20%, 40%, 60%, and 75%. Results for 0% mismatch are provided in Appendix B.1, with detailed class counts in Appendix C.1. *(ii)* For all SSL baselines, 40 labeled samples per known class are randomly selected, and the remaining known-class and selected unknown-class instances form the unlabeled set based on the mismatch proportion.

**Evaluations.** Our evaluation is conducted on the closed-set task and open-set task.

For the closed-set task, we report *known-class accuracy* on $K$-way classification, where test instances belong exclusively to known classes and are classified accordingly.

For the open-set task, leveraging Eq. (10), we evaluate a test set containing known, unknown, and new classes in a $K+1$-way classification setting, reporting: *(i)* known-class accuracy, reflecting correct classification of instances into their respective classes; *(ii)* unknown-class accuracy, measuring the assignment of instances from unknown classes to the unified "other" class; *(iii)* new-class accuracy, assessing generalization by categorizing instances from new classes to the unified "other" class; *(iv)* balance score, defined as the mean accuracy minus its standard deviation, which captures performance and volatility across these accuracies.

Unlike prior work (Saito et al., 2021; Li et al., 2023), which evaluates only unknown/new classes or reports average recall, we individually assess and report all three metrics and introduce the balance score to measure overall performance.

*Table 2.* The balance score (bala.) and average accuracy on known (kno.), unknown (unko.), and new classes for the open-set task on the CIFAR-10 dataset across mismatch proportions from 20% to 75%. A higher balance score indicates better and more balanced performance across the known, unknown, and new classes. The best and second-best results are highlighted in **bold** and underlined, respectively.

| method | 20% accuracy | | | 20% bala. | 40% accuracy | | | 40% bala. | 60% accuracy | | | 60% bala. | 75% accuracy | | | 75% bala. |
|---|---|---|---|---|---|---|---|---|---|---|---|---|---|---|---|---|
| | kno. | unkno. | new | | kno. | unkno. | new | | kno. | unkno. | new | | kno. | unkno. | new | |
| DS³L | 65.6 | 0.0 | 0.0 | -16.0 | 67.3 | 0.0 | 0.0 | -16.4 | 66.6 | 0.0 | 0.0 | -16.3 | 68.3 | 0.0 | 0.0 | -16.7 |
| UASD | 82.2 | 0.0 | 0.0 | -20.1 | 78.2 | 0.0 | 0.0 | -19.1 | 79.3 | 0.0 | 0.0 | -19.4 | 68.8 | 0.0 | 0.0 | -16.8 |
| CCSSL | **97.9** | 0.0 | 0.0 | -23.9 | **96.0** | 0.0 | 0.0 | -23.4 | **95.7** | 0.0 | 0.0 | -23.4 | 94.3 | 0.0 | 0.0 | -23.0 |
| T2T | - | - | - | - | - | - | - | - | - | - | - | - | - | - | 0.0 | - |
| MCTF | 54.7 | 0.0 | 0.0 | -13.3 | 62.4 | 0.0 | 0.0 | -15.2 | 83.2 | 0.0 | 0.0 | -20.3 | 71.8 | 0.0 | 0.0 | -17.5 |
| IOMatch | 96.2 | 1.6 | 6.0 | -18.8 | 91.5 | 3.2 | 6.1 | -16.6 | 87.5 | 7.7 | 6.3 | -12.6 | 84.1 | 7.3 | 6.0 | -12.3 |
| OpenMatch | 43.1 | 36.8 | 35.8 | 34.7 | 67.3 | 27.8 | 20.0 | 13.0 | 47.4 | 50.8 | 52.3 | 47.7 | 71.3 | 1.2 | 4.4 | -14.0 |
| Ours | 92.4 | **100.0** | **97.9** | **92.9** | 90.4 | **100.0** | **99.8** | **91.2** | 94.0 | **100.0** | **100.0** | **94.6** | 95.8 | **100.0** | **99.1** | **96.1** |

*T2T is excluded as it is not applicable to binary classification.

*Table 3.* The balance score (bala.) and average accuracy on known (kno.), unknown (unko.), and new classes for the open-set task on the CIFAR-100 across mismatch proportions from 20% to 75%. A higher balance score indicates better and more balanced performance across the known, unknown, and new classes. The best and second-best results are highlighted in **bold** and underlined, respectively.

| method | 20% accuracy | | | 20% bala. | 40% accuracy | | | 40% bala. | 60% accuracy | | | 60% bala. | 75% accuracy | | | 75% bala. |
|---|---|---|---|---|---|---|---|---|---|---|---|---|---|---|---|---|
| | kno. | unkno. | new | | kno. | unkno. | new | | kno. | unkno. | new | | kno. | unkno. | new | |
| DS³L | 23.9 | 0.0 | 0.0 | -5.8 | 22.7 | 0.0 | 0.0 | -5.5 | 23.4 | 0.0 | 0.0 | -5.7 | 24.5 | 0.0 | 0.0 | -6.0 |
| UASD | 26.2 | 0.0 | 0.0 | -6.4 | 24.4 | 0.0 | 0.0 | -6.0 | 22.8 | 0.0 | 0.0 | -5.6 | 20.4 | 0.0 | 0.0 | -5.0 |
| CCSSL | 48.9 | 0.0 | 0.0 | -11.9 | 47.9 | 0.0 | 0.0 | -11.7 | 45.6 | 0.0 | 0.0 | -11.1 | 46.0 | 0.0 | 0.0 | -11.2 |
| T2T | **54.7** | 0.0 | 0.0 | -13.4 | **53.9** | 0.0 | 0.0 | -13.2 | **50.6** | 0.0 | 0.0 | -12.4 | 48.7 | 0.0 | 0.0 | -11.9 |
| MCTF | 0.0 | 98.7 | 98.8 | 8.8 | 8.9 | 35.8 | 34.9 | 11.3 | 40.2 | 0.9 | 0.7 | -8.8 | **56.6** | 0.0 | 0.0 | -13.8 |
| IOMatch | 0.0 | **100.0** | **100.0** | 9.0 | 0.0 | 100.0 | 100.0 | 9.0 | 0.0 | **100.0** | **100.0** | 9.0 | 0.0 | **100.0** | **100.0** | 9.0 |
| OpenMatch | 15.6 | 18.5 | 17.8 | 15.8 | 10.4 | 7.8 | 7.6 | 7.1 | 10.0 | 6.7 | 6.6 | 5.9 | 4.4 | 33.5 | 33.5 | 7.0 |
| Ours | 40.0 | 94.8 | 94.6 | **44.8** | 39.3 | 86.0 | 90.1 | **43.6** | 45.1 | 70.4 | 79.1 | **47.2** | 44.5 | 74.6 | 75.9 | **47.2** |

**Baseline methods.** We evaluate our approach against four SCDM$_{CT}$ methods: UASD (Chen et al., 2020), DS³L (Guo et al., 2020), T2T (Huang et al., 2021), and CCSSL (Yang et al., 2022), and three SCDM methods: MTCF (Yu et al., 2020), OpenMatch (Saito et al., 2021), and IOMatch (Li et al., 2023).

**Implementation Details.** All experiments utilize the pre-trained Stable Diffusion 2.0 model (Rombach et al., 2022) as the DPM generator, without further optimization. Following (Chen et al., 2020; Guo et al., 2020; Saito et al., 2021), the classifier adopts the WideResNet-28-2 (Zagoruyko & Komodakis, 2016) backbone. Each method is run three times per dataset, and the mean accuracy is reported. For more details, please refer to Appendix C.2. The code is available at `https://github.com/RUC-DWBI-ML/research/tree/main/UCDM-master`.

### 4.2. Experimental Results

CIFAR-10 includes 2 known, 6 unknown, and 2 new classes. CIFAR-100 is harder with 20 known, 60 unknown, and 20 new. Tiny-ImageNet is the most complex, with 20 known, 80 unknown, and 100 new classes. The highest and second-highest accuracies, along with the balance score, are **bolded**

and underlined, respectively.

**Performance on closed-set task.** Tab. 1 shows the closed-set task results on CIFAR-10, CIFAR-100, and Tiny-ImageNet across mismatch proportions from 20% to 75%. From the results, we have following two key observations. *(i)* UCDM achieves the second-highest accuracy at least twice on each dataset and outperforms all compared methods on CIFAR-10 at a 75% mismatch proportion, highlighting its ability to train an effective closed-set classifier without relying on ground truth labels. *(ii)* As the mismatch proportion increases, UCDM consistently improves, demonstrating its robustness across varying mismatch proportions.

**Performance on open-set task.** Tab. 2, Tab. 3, and Tab. 4 present open-set results, including balance scores and accuracies for known, unknown, and new classes on CIFAR-10, CIFAR-100, and Tiny-ImageNet. From the results, we observe that UCDM achieves the highest balance score across all settings in the three datasets. This demonstrates its capability to maintain high mean accuracy and low standard deviation across known, unknown, and new classes, even on the more challenging Tiny-ImageNet benchmark.

In contrast, SCDM$_{CT}$ methods fail to classify unknown and

*Table 4.* The balance score (bala.) and average accuracy on known (kno.), unknown (unko.), and new classes for the open-set task on the Tiny-ImageNet across mismatch proportions from 20% to 75%. A higher balance score indicates better and more balanced performance across the known, unknown, and new classes. The best and second-best results are highlighted in **bold** and underlined, respectively.

| method | 20% accuracy | | | bala. | 40% accuracy | | | bala. | 60% accuracy | | | bala. | 75% accuracy | | | bala. |
|---|---|---|---|---|---|---|---|---|---|---|---|---|---|---|---|---|
| | kno. | unkno. | new | | kno. | unkno. | new | | kno. | unkno. | new | | kno. | unkno. | new | |
| DS$^3$L | 24.5 | 0.0 | 0.0 | -6.0 | 25.7 | 0.0 | 0.0 | -6.3 | 26.3 | 0.0 | 0.0 | -6.4 | 25.7 | 0.0 | 0.0 | -6.3 |
| UASD | 5.3 | 0.0 | 0.0 | -1.3 | 6.8 | 0.0 | 0.0 | -1.7 | 5.2 | 0.0 | 0.0 | -1.3 | 6.7 | 0.0 | 0.0 | -1.6 |
| CCSSL | 26.7 | 0.0 | 0.0 | -6.5 | 23.4 | 0.0 | 0.0 | -5.7 | 25.8 | 0.0 | 0.0 | -6.3 | 24.4 | 0.0 | 0.0 | -6.0 |
| T2T | **40.5** | 0.0 | 0.0 | -9.9 | **41.0** | 0.0 | 0.0 | -10.0 | **41.7** | 0.0 | 0.0 | -10.2 | **38.0** | 0.0 | 0.0 | -9.3 |
| MCTF | 0.9 | 4.5 | 92.6 | -19.3 | 19.1 | 0.9 | 8.1 | 0.2 | 24.2 | 0.1 | 0.1 | -5.8 | 26.6 | 0.0 | 0.0 | -6.5 |
| IOMatch | 0.0 | **100.0** | **100.0** | 8.9 | 0.0 | **100.0** | **100.0** | 8.9 | 0.0 | **100.0** | **100.0** | 8.9 | 0.0 | **100.0** | **100.0** | 8.9 |
| OpenMatch | 13.3 | 20.6 | 24.0 | 13.8 | 9.8 | 22.4 | 23.6 | 11.0 | 10.8 | 3.5 | 5.9 | 3.0 | 10.8 | 35.2 | 33.7 | 12.9 |
| Ours | 21.9 | 86.8 | 86.7 | **27.6** | 17.3 | 95.3 | 94.6 | **24.2** | 15.8 | 94.9 | 95.4 | **22.9** | 16.8 | 88.5 | 88.7 | **23.2** |

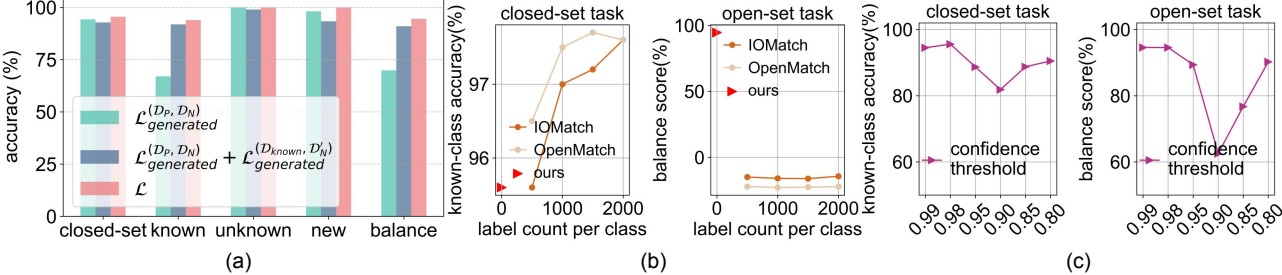

*Figure 4.* Ablation studies: (a) shows the ablation study on learning objectives, demonstrating the effectiveness of each component. (b) compares our method with SSL across varying label counts, highlighting its cost-saving potential. (c) analyzes the sensitivity to the confidence threshold, suggesting a higher threshold for stable performance.

new classes into a unified "other" category, resulting in zero accuracy for these tasks. Meanwhile, SCDM methods exhibit low balance scores due to either low mean accuracy or high standard deviation when classifying known, unknown, and new classes. UCDM, however, demonstrates robustness across varying mismatch proportions and datasets.

### 4.3. Ablation Studies

**Learning objectives.** Fig. 4 (a) evaluates loss components on CIFAR-10 (60% mismatch), reporting known-class accuracy for the closed-set task, and balance score and accuracies for known, unknown, and new classes in the open-set task.

The results reveal two key findings: *(i)* UCDM achieves the best results across all evaluation criteria when optimized with the full loss component $\mathcal{L}$, demonstrating its effectiveness. *(ii)* Realistic images play a crucial role in improving the classification performance of known classes. For instance, when comparing the framework optimized with $\mathcal{L}_{\text{generated}}^{(\mathcal{D}_P, \mathcal{D}_N)} + \mathcal{L}_{\text{generated}}^{(\mathcal{D}_{\text{known}}, \mathcal{D}'_N)}$ to $\mathcal{L}_{\text{generated}}^{(\mathcal{D}_P, \mathcal{D}_N)}$, there is a notable improvement in performance for known classes.

**Effectiveness of positive-negative instance generation.** Tab. 5 shows the comparison between instance generation from random noise and our method.

*Table 5.* Compare our pipeline with random noise-based instances on CIFAR-10 (60% mismatch), reporting known-class accuracy (closed-set) and balance score (open-set). $\text{UCDM}_{p \backslash n}$ use our positive\negative pipeline if selected; otherwise, random noise is applied. Best and second-best results are **bolded** and underlined.

| variants | $\text{UCDM}_p$ | $\text{UCDM}_n$ | known-class | balance |
|---|---|---|---|---|
| I | ✔ | ✔ | **95.6** | **94.6** |
| II | ✗ | ✔ | 82.9 | 72.8 |
| III | ✔ | ✗ | 90.0 | 79.3 |
| IV | ✗ | ✗ | 84.8 | 71.5 |

The results highlight two key observations: *(i)* Generating positive instances with our pipeline yields the best (I) and second-best (III) performance, demonstrating its effectiveness. *(ii)* Randomly generating positive or negative instances significantly reduces the balance score, as seen in II, III, and IV, due to the lack of effective comparisons.

**Comparison with SSL under varying label counts per class.** Fig. 4 (b) compares our method with SCDM methods like IOMatch and OpenMatch using varying label counts per class on CIFAR-10 with a 60% mismatch proportion.

The results show that UCDM, without labels, outperforms IOMatch and OpenMatch (with 2,000 labels per class) in balance score. Additionally, it achieves superior known-class

accuracy on the closed-set task compared to OpenMatch with 500 labels per class. These findings highlight UCDM's effectiveness and its ability to reduce annotation costs.

**Confidence threshold.** Fig. 4 (c) analyzes the sensitivity of the confidence threshold in confidence-based labeling. The results show that the model remains robust with a threshold above 0.95. However, performance, especially in the open-set task, declines below 0.95 due to incorrect pseudo-labels. We recommend using a higher threshold (e.g., $\delta = 0.98$, as in our experiments) to ensure stable performance.

## 5. Conclusion

Previous studies on CDM focus on SCDM$_{CT}$ and SCDM, where the reliance on labeled data limits their applicability to unsupervised scenarios and hinders performance in open-set tasks. To overcome this, we propose Unsupervised Learning for Class Distribution Mismatch (UCDM), which uses a diffusion model to create or erase semantic classes in unlabeled images, generating positive and negative pairs for classifier training. We also provide two theorems to theoretically support this approach. This framework mitigates the need for ground truth labels and extends applicability to unsupervised settings. Extensive experiments on various tasks and datasets show UCDM's superior performance.

**Limitations and future work** Limited prompt variability restricts positive instance diversity in UCDM. Integrating a large language model could improve this.

## Acknowledgements

This work is supported by the National Key Research & Develop Plan(2023YFB4503600), National Natural Science Foundation of China(U23A20299, U24B20144, 62276270, 62322214, 62172424, 62076245), Beijing Natural Science Foundation(4212022), Program of China Scholarship Council, and the Outstanding Innovative Talents Cultivation Funded Programs 2024 of Renmin University of China. Meanwhile, it is supported by the NUS startup grant (Presidential Young Professorship), Singapore MOE Tier-1 grant, ByteDance grant, ARCTIC grant, SMI grant (WBS number: A8001104-00-00), Alibaba grant, and Google grant for TPU usage. It is also partially supported by the Opening Fund of Hebei Key Laboratory of Machine Learning and Computational Intelligence.

## Impact Statement

This paper advances the field of machine learning with a particular focus on unsupervised class distribution mismatch. While the work carries many potential societal implications, none are deemed necessary to emphasize explicitly here.

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

We organize our appendix as follows.

**Algorithm:**

- Loss function and pseudo-code

  - Section A.1: Details of loss function
  - Section A.2: Details on constructing pairs of real and generated images
  - Section A.3: Pseudo-Code for diffusion-driven data generation and classifier training

- Proof

  - Section A.4: Empirical evidence of negligible $\delta_t$ and $\tilde{\delta}_t$
  - Section A.5: Proof of Theorem 3.1
  - Section A.6: Proof of the forward process in negative instance generation
  - Section A.7: Proof of Theorem 3.2

**Additional experimental results:**

- Mismatch settings

  - Section B.1: Experimental results on 0% mismatch proportion across different datasets
  - Section B.2: Experimental results on categories with varying proportions

- Sensitive and ablation analysis

  - Section B.3: Experimental results on generated positive instances with varying parameter $\sigma_t$
  - Section B.4: Experimental results on generated negative instances with varying parameter $\sigma_t$
  - Section B.5: Analysis of the sensitivity to weights in the loss function
  - Section B.6: Evaluation of the impact of the batch normalization layer on model training
  - Section B.7: Evaluation of pseudo-label reliability in selected instances

- Visualization

  - Section B.8: Visualization of generated positive images from random noise
  - Section B.9: Visualization of generated negative images from random noise
  - Section B.10: Visualization of generated positive images with varying parameter $\sigma_t$
  - Section B.11: Visualization of generated negative images with varying parameter $\sigma_t$
  - Section B.12: Visualization of DDIM Inversion and Negative Instances Generation in UCMD
  - Section B.13: Visualization of generated positive and negative images in UCDM
  - Section B.14: Visualization of generated harder training pairs

**Experimental settings:**

- Section C.1: Dataset details

- Section C.2: Training details

# A. Algorithm

## A.1. Details of Loss Function

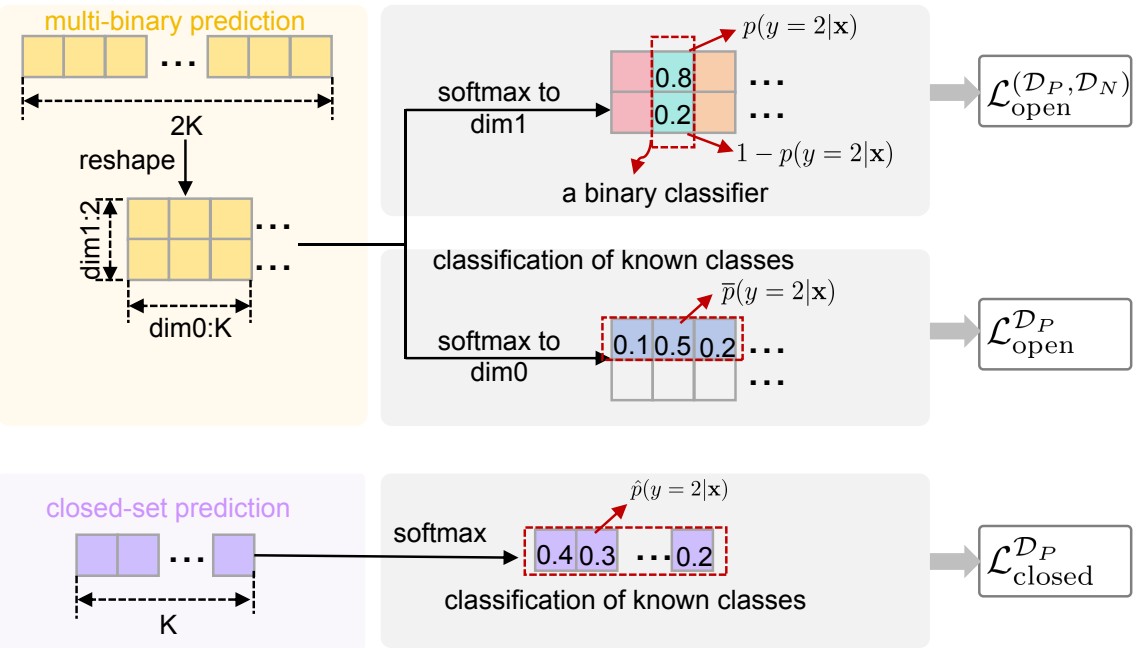

*Figure 5.* Schematic diagram of loss functions.

To provide a clear understanding of the loss computation mechanism, we illustrates the process of deriving the loss function from the logits in Fig. 5.

For the open-set task, we train an open-set classifier using both the positive and negative instance sets with the loss function $\mathcal{L}_{\text{open}}^{(\mathcal{D}_P, \mathcal{D}_N)}$, which aims to maximize the probability of positive instances being correctly assigned to their respective classes, i.e., $p(y|\boldsymbol{x})$, while minimizing the probability of assigning them to their corresponding negative counterparts, as shown in Eq. (13).

$$\mathcal{L}_{\text{open}}^{(\mathcal{D}_P, \mathcal{D}_N)} = \frac{1}{|\mathcal{D}_P|} \sum_{(\boldsymbol{x},y)\in\mathcal{D}_P} -\log p(y|\boldsymbol{x}) + \frac{1}{|\mathcal{D}_N|} \sum_{(\boldsymbol{x},y)\in\mathcal{D}_N} -\log\left[1 - p(y|\boldsymbol{x})\right]. \tag{13}$$

Simultaneously, to improve the identification of known classes, we impose a constraint on the multi-binary prediction for dimension 0, ensuring that the probability corresponding to the ground truth label is maximized, as shown in Eq. (14).

$$\mathcal{L}_{\text{open}}^{(\mathcal{D}_P)} = -\frac{1}{|\mathcal{D}_P|} \sum_{(\boldsymbol{x},y)\in\mathcal{D}_P} \log \overline{p}(y|\boldsymbol{x}), \tag{14}$$

Additionally, for the closed-set task, we train a closed-set classifier by minimizing the loss function Eq. (15).

$$\mathcal{L}_{\text{closed}}^{(\mathcal{D}_P)} = -\frac{1}{|\mathcal{D}_P|} \sum_{(\boldsymbol{x},y)\in\mathcal{D}_P} \log \hat{p}(y|\boldsymbol{x}), \tag{15}$$

Consequently, the loss function for the generated dataset is defined as:

$$\mathcal{L}_{\text{generated}}^{(\mathcal{D}_P, \mathcal{D}_N)} = \lambda_1 \mathcal{L}_{\text{open}}^{(\mathcal{D}_P, \mathcal{D}_N)} + \lambda_2 \left[\mathcal{L}_{\text{open}}^{(\mathcal{D}_P)} + \mathcal{L}_{\text{closed}}^{(\mathcal{D}_P)}\right],$$

where $\lambda_1$ and $\lambda_2$ control the trade-off for each objective.

### A.2. Details on constructing pairs of real and generated images.

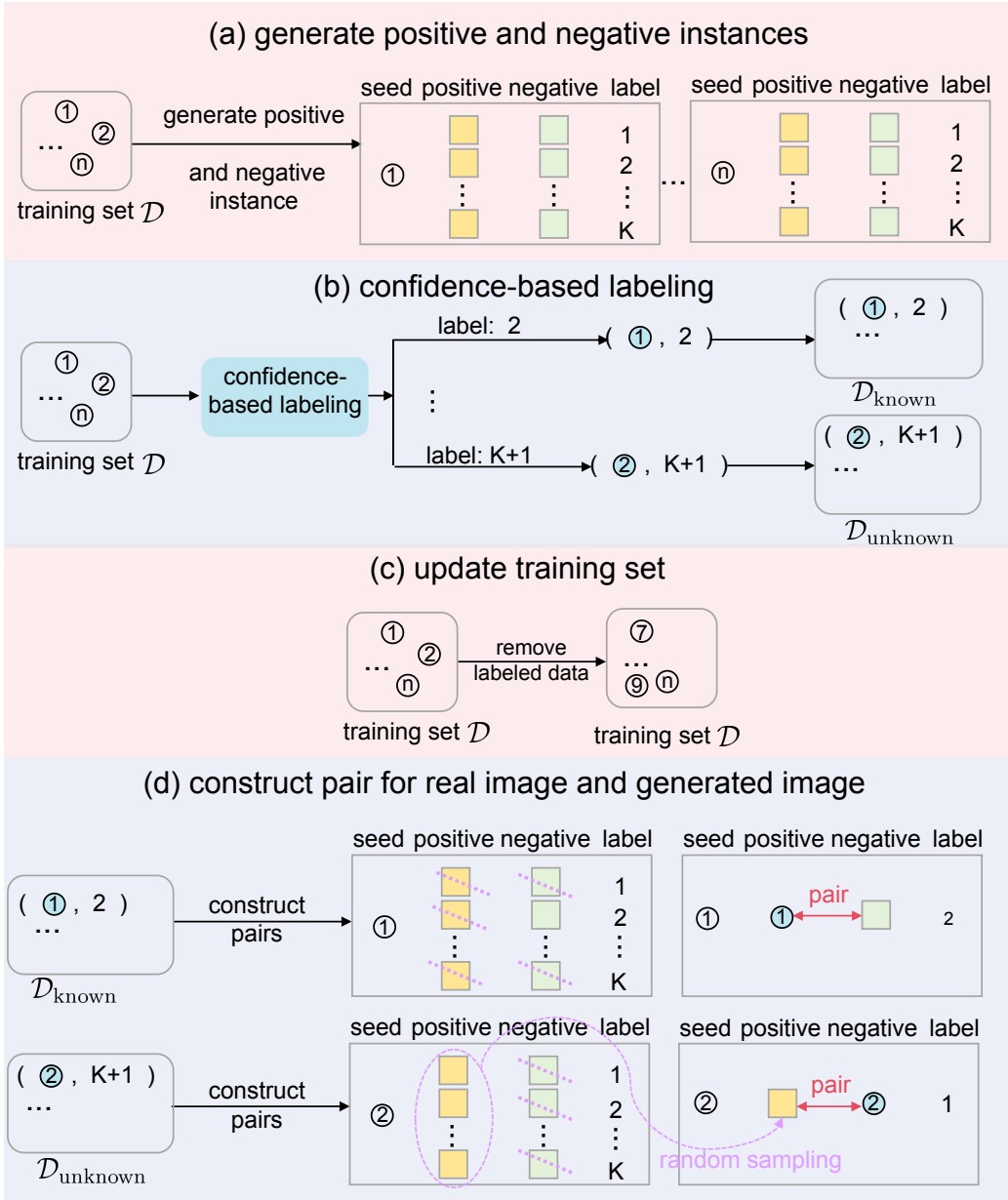

*Figure 6.* Schematic diagram for constructing positive and negative pairs.

In Fig. 6, we present a toy example to visualize the process of constructing pairs.

In the first step, as shown in Fig. 6 (a), we generate $K$ positive and negative instances for each seed sample from the training set, where $K$ denotes the number of known classes. Each positive and negative instance derived from the same seed sample forms a pair.

In the second step, as shown in Fig. 6 (b), we adopt a confidence-based labeling mechanism to assign pseudo-labels to high-confidence samples. These samples are then removed from the training set, as depicted in Fig. 6 (c). Instances with

pseudo-labels corresponding to known classes form the set $\mathcal{D}_{\text{known}}$, while those with the pseudo-label $K + 1$ are placed in $\mathcal{D}_{\text{unknown}}$.

Finally, we aim to construct pairs for real images from $\mathcal{D}_{\text{known}}$ and $\mathcal{D}_{\text{unknown}}$, as shown in Fig. 6 (d). For an instance $x$ from $\mathcal{D}_{\text{known}}$, we select a generated negative instance based on the seed sample $x$ that has the same pseudo-label as $x$ to serve as its negative counterpart. For an instance $x$ from $\mathcal{D}_{\text{unknown}}$, we randomly sample a generated positive instance corresponding to $x$ to serve as its positive counterpart. Meanwhile, the remaining generated instances based on $x$ do not participate in training.

### A.3. Pseudo-Code for Diffusion-Driven Data Generation and Classifier Training

To facilitate a better understanding of our problem setup and proposed method UCDM, we provide the pseudo code below.

---

**Algorithm 1** Diffusion-based data generation

---

  # Sample generation stage
**Input:** training set $\mathcal{D}$, the prompt set of known classes $\mathcal{C}$, diffusion model, positive instance set $\mathcal{D}_P$, negative instance set $\mathcal{D}_N$
Initialize $\mathcal{D}_P = \emptyset$, $\mathcal{D}_N = \emptyset$
**for** $x$ **in** $\mathcal{D}$ **do**
  **for** $\mathcal{C}_y$ **in** $\mathcal{C}$ **do**
    Forward $x$ to noise vectors $\hat{x}_T$ and $x_T$ using Eq. (1) and Eq. (6), respectively.
    Forward $\hat{x}_T$ and $\mathcal{C}_y$ to the diffusion model to obtain $\hat{x}_0$ using Eq. (2), and add $\hat{x}_0$ to $\mathcal{D}_P$.
    Forward $x_T$ to the diffusion model to obtain $\tilde{x}_0$ using Eq. (8), and add $\tilde{x}_0$ to $\mathcal{D}_N$.
  **end for**
**end for**

---

---

**Algorithm 2** UCDM: **U**nsupervised Learning for **C**lass **D**istribution **M**ismatch

---

# Training classifier stage

**Input:** Training set $\mathcal{D}$, positive instance set $\mathcal{D}_P$, negative instance set $\mathcal{D}_N$, set of real instances with pseudo-labels from known classes $\mathcal{D}_{\text{known}}$, set of real instances with pseudo-labels from unknown classes $\mathcal{D}_{\text{unknown}}$, known-class set $\mathcal{Y}_{\text{known}}$, negative instances of $\mathcal{D}_{\text{known}}$: $\mathcal{D}'_N$ , positive instances of $\mathcal{D}_{\text{unknown}}$: $\mathcal{D}'_P$, confidence-based labeling epoch $e_t$, classifier

Initialize $\mathcal{D}_{\text{known}} = \emptyset, \mathcal{D}_{\text{unknown}} = \emptyset, \mathcal{D}'_P = \emptyset, \mathcal{D}'_N = \emptyset$

**for** epoch = 1, 2, ... **do**

  **for** $\mathcal{S}$ **in** $\{\mathcal{D}, \mathcal{D}_{\text{known}}, \mathcal{D}_{\text{unknown}}\}$ **do**

    **if** $\mathcal{S} \neq \emptyset$ **then**

      Sample $\boldsymbol{x}$ from $\mathcal{S}$.

      **if** $\mathcal{S} = \mathcal{D}$ **then**

        Sample $\mathcal{B}_p$ and $\mathcal{B}_n$ from $\mathcal{D}_P$ and $\mathcal{D}_N$, where $\mathcal{B}_p$ and $\mathcal{B}_n$ indicate the generated positive and negative instance set based on $\boldsymbol{x}$ and prompt set $\mathcal{C}$.

      **else if** $\mathcal{S} = \mathcal{D}_{\text{known}}$ **then**

        Sample $\tilde{\boldsymbol{x}}$ from $\mathcal{D}'_N$ as a negative instance, where $\tilde{\boldsymbol{x}}$ is generated based on $\boldsymbol{x}$ and $\mathcal{C}_y$, with $y$ being the pseudo-label of $\boldsymbol{x}$.

      **else if** $\mathcal{S} = \mathcal{D}_{\text{unknown}}$ **then**

        Sample $\hat{\boldsymbol{x}}$ from $\mathcal{D}'_P$ as a positive instance, where $\hat{\boldsymbol{x}}$ is generated based on $\boldsymbol{x}$ and $\mathcal{C}_y$, with $y$ being randomly sampled from $\{1, 2, \ldots, K\}$.

      **end if**

      Train classifier with sampled data using Eq. (12).

    **end if**

  **end for**

  # Confidence-based labeling

  **if** epoch = $e_c$ **then**

    **for** $\boldsymbol{x}$ **in** $\mathcal{D}$ **do**

      Forward $\boldsymbol{x}$ to the classifier to obtain $\boldsymbol{q}$ and $\tilde{\boldsymbol{q}}$ using Eq. (10) and Eq. (11).

      **if** $\arg\max \boldsymbol{q} = \arg\max \tilde{\boldsymbol{q}}$ **and** $\max \boldsymbol{q} \geq \delta$ **and** $\max \tilde{\boldsymbol{q}} \geq \delta$ **then**

        Assign pseudo-label $\arg\max \boldsymbol{q}$ to $\boldsymbol{x}$.

        **if** $\arg\max \boldsymbol{q} \in \mathcal{Y}_{\text{known}}$ **then**

          Add $(\boldsymbol{x}, \arg\max \boldsymbol{q})$ to $\mathcal{D}_{\text{known}}$, and select negatives from $\mathcal{D}_N$ to add to $\mathcal{D}'_N$.

        **else**

          Add $(\boldsymbol{x}, \arg\max \boldsymbol{q})$ to $\mathcal{D}_{\text{unknown}}$, and select positives from $\mathcal{D}_P$ to add to $\mathcal{D}'_P$.

        **end if**

        Remove $\boldsymbol{x}$ from $\mathcal{D}$ and corresponding instances from $\mathcal{D}_P$ and $\mathcal{D}_N$.

      **end if**

    **end for**

  **end if**

**end for**

**Return:** classifier

---

### A.4. Empirical evidence of negligible $\delta_t$ and $\tilde{\delta}_t$

To assess the negligibility of $\delta_t$ and $\tilde{\delta}_t$, we compute the 1-cosine similarity between $\epsilon(\mathbf{x}_t, t, \mathcal{C}_y)$ and $\epsilon(\mathbf{x}_{t-1}, t, \mathcal{C}_y)$, as well as between $\epsilon(\mathbf{x}_t, t)$ and $\epsilon(\mathbf{x}_{t-1}, t)$, over 20 DDIM steps. As shown in Tab. 6, the results indicate near-perfect alignment, validating that both $\delta_t$ and $\tilde{\delta}_t$ are negligible in practice.

Table 6. Values of $\delta_t$ and $\tilde{\delta}_t$ across diffusion steps for condition and uncondition settings.

| Step | 1 | 2 | 3 | 4 | 5 | 6 | 7 | 8 | 9 | 10 | 11 | 12 | 13 | 14 | 15 | 16 | 17 | 18 | 19 | 20 |
|---|---|---|---|---|---|---|---|---|---|---|---|---|---|---|---|---|---|---|---|---|
| $\delta_t$ | 4e-2 | 3e-2 | 2e-2 | 2e-2 | 1e-2 | 8e-3 | 5e-3 | 3e-3 | 3e-3 | 2e-3 | 2e-3 | 2e-3 | 2e-3 | 2e-3 | 1e-3 | 1e-3 | 1e-3 | 1e-3 | 1e-3 | 4e-4 |
| $\tilde{\delta}_t$ | 4e-2 | 3e-2 | 2e-2 | 2e-2 | 1e-2 | 8e-3 | 5e-3 | 3e-3 | 3e-3 | 3e-3 | 2e-3 | 2e-3 | 2e-3 | 2e-3 | 1e-3 | 1e-3 | 1e-3 | 1e-3 | 1e-3 | 4e-4 |

### A.5. Proof of Theorem 3.1

*Proof.* According to Eq. (2), when $\gamma = 1$ and $\sigma_t = 0$, we have the following formula:

$$\boldsymbol{x}_{t-1} = \sqrt{\frac{\alpha_{t-1}}{\alpha_t}}\boldsymbol{x}_t - \sqrt{\alpha_{t-1}}\psi(\alpha_t, \alpha_{t-1}, 0)\epsilon_\theta(\boldsymbol{x}_t, t, \mathcal{C}_{\boldsymbol{y}})$$

We can then represent $\boldsymbol{x}_t$ as:

$$
\begin{aligned}
\boldsymbol{x}_t &= \sqrt{\frac{\alpha_t}{\alpha_{t-1}}}\boldsymbol{x}_{t-1} + \sqrt{\alpha_t}\left(\sqrt{\frac{1}{\alpha_t}-1} - \sqrt{\frac{1}{\alpha_{t-1}}-1}\right)\epsilon_\theta(\boldsymbol{x}_t, t, \mathcal{C}_{\boldsymbol{y}}) \\
&= \sqrt{\frac{\alpha_t}{\alpha_{t-1}}}\left(\sqrt{\frac{\alpha_{t-1}}{\alpha_{t-2}}}\boldsymbol{x}_{t-2} + \sqrt{\alpha_{t-1}}\left(\sqrt{\frac{1}{\alpha_{t-1}}-1} - \sqrt{\frac{1}{\alpha_{t-2}}-1}\right)\epsilon_\theta(\boldsymbol{x}_{t-1}, t-1, \mathcal{C}_{\boldsymbol{y}})\right) \\
&\quad + \sqrt{\alpha_t}\left(\sqrt{\frac{1}{\alpha_t}-1} - \sqrt{\frac{1}{\alpha_{t-1}}-1}\right)\epsilon_\theta(\boldsymbol{x}_t, t, \mathcal{C}_{\boldsymbol{y}}) \\
&= \sqrt{\frac{\alpha_t}{\alpha_{t-1}}}\left(\sqrt{\frac{\alpha_{t-1}}{\alpha_{t-2}}}\left[\sqrt{\frac{\alpha_{t-2}}{\alpha_{t-3}}}\boldsymbol{x}_{t-3} + \sqrt{\alpha_{t-2}}\left(\sqrt{\frac{1}{\alpha_{t-2}}-1} - \sqrt{\frac{1}{\alpha_{t-3}}-1}\right)\epsilon_\theta(\boldsymbol{x}_{t-2}, t-2, \mathcal{C}_{\boldsymbol{y}})\right]\right. \\
&\quad \left. + \sqrt{\alpha_{t-1}}\left(\sqrt{\frac{1}{\alpha_{t-1}}-1} - \sqrt{\frac{1}{\alpha_{t-2}}-1}\right)\right)\epsilon_\theta(\boldsymbol{x}_{t-1}, t-1, \mathcal{C}_{\boldsymbol{y}}) \\
&\quad + \sqrt{\alpha_t}\left(\sqrt{\frac{1}{\alpha_t}-1} - \sqrt{\frac{1}{\alpha_{t-1}}-1}\right)\epsilon_\theta(\boldsymbol{x}_t, t, \mathcal{C}_{\boldsymbol{y}}) \\
&= \sqrt{\frac{\alpha_t\alpha_{t-1}\ldots\alpha_1}{\alpha_{t-1}\alpha_{t-2}\ldots\alpha_0}}\boldsymbol{x}_0 + \sum_{i=1}^t \sqrt{\alpha_t}\left(\sqrt{\frac{1}{\alpha_i}-1} - \sqrt{\frac{1}{\alpha_{i-1}}-1}\right)\epsilon_\theta(\boldsymbol{x}_i, i, \mathcal{C}_{\boldsymbol{y}}).
\end{aligned}
\tag{16}
$$

Let $\epsilon_\theta(\boldsymbol{x}_i, i, \mathcal{C}_{\boldsymbol{y}}) - \epsilon_\theta(\boldsymbol{x}_{i-1}, i, \mathcal{C}_{\boldsymbol{y}}) = \delta_i$. Then, we have

$$\epsilon_\theta(\boldsymbol{x}_t, t, \mathcal{C}_{\boldsymbol{y}}) = -\sqrt{1-\bar{\alpha}_t}\nabla\log p_\theta(\boldsymbol{x}_{t-1} \mid y) + \delta_t,$$

which leads to:

$$
\begin{aligned}
\boldsymbol{x}_t &= \sqrt{\frac{\alpha_t\alpha_{t-1}\ldots\alpha_1}{\alpha_{t-1}\alpha_{t-2}\ldots\alpha_0}}\boldsymbol{x}_0 - \sum_{i=0}^{t-1}\sqrt{\alpha_t(1-\bar{\alpha}_{i+1})}\left(\sqrt{\frac{1}{\alpha_{i+1}}-1} - \sqrt{\frac{1}{\alpha_i}-1}\right)\nabla_{\boldsymbol{x}_i}\log p_\theta(\boldsymbol{x}_i \mid y) \\
&\quad + \sum_{i=0}^{t-1}\sqrt{\alpha_t}\left(\sqrt{\frac{1}{\alpha_{i+1}}-1} - \sqrt{\frac{1}{\alpha_i}-1}\right)\delta_{i+1}.
\end{aligned}
\tag{17}
$$

Let $s_i = \sqrt{\alpha_t(1 - \bar{\alpha}_{i+1})}\left(\sqrt{\frac{1}{\alpha_{i+1}} - 1} - \sqrt{\frac{1}{\alpha_i} - 1}\right)$, for $0 \leq i \leq t - 1$. Then the above expression simplifies to:

$$\boldsymbol{x}_t = \sqrt{\alpha_t}\boldsymbol{x}_0 - \sum_{i=0}^{t-1} s_i \nabla_{\boldsymbol{x}_i} \log p_\theta(\boldsymbol{x}_i \mid y) + \sum_{i=0}^{t-1} \frac{s_i}{1 - \sqrt{\bar{\alpha}_{i+1}}}\delta_{i+1}. \tag{18}$$

Applying Bayes' theorem, the conditional term can be rewritten as:

$$\boldsymbol{x}_t = \sqrt{\alpha_t}\boldsymbol{x}_0 - \sum_{i=0}^{t-1} \nabla_{\boldsymbol{x}_i} \log \left(\frac{p_\theta(\boldsymbol{x}_i)p_\theta(y \mid \boldsymbol{x}_i)}{p_\theta(y)}\right)^{s_i} + \sum_{i=0}^{t-1} \frac{s_i}{1 - \sqrt{\bar{\alpha}_{i+1}}}\delta_{i+1}. \tag{19}$$

Since the gradient of $\log p_\theta(y)$ with respect to $\boldsymbol{x}_i$ is zero, we obtain:

$$\boldsymbol{x}_t = \sqrt{\alpha_t}\boldsymbol{x}_0 - \sum_{i=0}^{t-1} [\nabla_{\boldsymbol{x}_i} \log p_\theta(\boldsymbol{x}_i)^{s_i} + \nabla_{\boldsymbol{x}_i} \log p_\theta(y \mid \boldsymbol{x}_i)^{s_i}] + \sum_{i=0}^{t-1} \frac{s_i}{1 - \sqrt{\bar{\alpha}_{i+1}}}\delta_{i+1}. \tag{20}$$

The proof is complete. □

### A.6. Proof of the Forward Process in Negative Instance Generation

*Proof.* From Eq. (2), when $\gamma = 1$ and $\sigma_t = 0$, the following equation holds:

$$\boldsymbol{x}_{t-1} = \sqrt{\frac{\alpha_{t-1}}{\alpha_t}}\boldsymbol{x}_t - \sqrt{\alpha_{t-1}}\psi(\alpha_t, \alpha_{t-1}, 0)\epsilon_\theta(\boldsymbol{x}_t, t, \mathcal{C}_{\boldsymbol{y}}).$$

Thus, we can represent $\boldsymbol{x}_t$ as:

$$\boldsymbol{x}_t = \sqrt{\frac{\alpha_t}{\alpha_{t-1}}}\boldsymbol{x}_{t-1} + \sqrt{\alpha_t}\psi(\alpha_t, \alpha_{t-1}, 0)\epsilon_\theta(\boldsymbol{x}_t, t, \mathcal{C}_{\boldsymbol{y}}).$$

Since $\epsilon_\theta(\boldsymbol{x}_t, t, \mathcal{C}_{\boldsymbol{y}})$ is not directly accessible, we adopt a forward Euler approximation, replacing $\epsilon_\theta(\boldsymbol{x}_t, t, \mathcal{C}_{\boldsymbol{y}})$ with $\epsilon_\theta(\boldsymbol{x}_{t-1}, t, \mathcal{C}_{\boldsymbol{y}})$ following DDIM (Song et al., 2020a). As a result, we obtain:

$$\boldsymbol{x}_t = \sqrt{\frac{\alpha_t}{\alpha t - 1}}\boldsymbol{x}_{t-1} + \sqrt{\alpha_t}\psi(\alpha_t, \alpha_{t-1}, 0), \epsilon_\theta(\boldsymbol{x}_{t-1}, t, \mathcal{C}_{\boldsymbol{y}}).$$

The proof is complete. □

### A.7. Proof of Theorem 3.2

*Proof.* We begin with the following equation:

$$\tilde{\boldsymbol{x}}_{t-1} = \sqrt{\frac{\alpha_{t-1}}{\alpha_t}}\tilde{\boldsymbol{x}}_t - \sqrt{\alpha_{t-1}}\left(\sqrt{\frac{1}{\alpha_t} - 1} - \sqrt{\frac{1}{\alpha_{t-1}} - 1}\right)\epsilon_\theta(\tilde{\boldsymbol{x}}_t, t) \tag{21}$$

Rearranging, we obtain:

$$\tilde{\boldsymbol{x}}_t = \sqrt{\frac{\alpha_t}{\alpha_{t-1}}}\tilde{\boldsymbol{x}}_{t-1} + \sqrt{\alpha_t}\left(\sqrt{\frac{1}{\alpha_t} - 1} - \sqrt{\frac{1}{\alpha_{t-1}} - 1}\right)\epsilon_\theta(\tilde{\boldsymbol{x}}_t, t)$$

Following a similar derivation process as in Eq. (16), we obtain Eq. (22):

$$\tilde{\boldsymbol{x}}_t = \sqrt{\frac{\alpha_t\alpha_{t-1}\dots\alpha_1}{\alpha_{t-1}\alpha_{t-2}\dots\alpha_0}}\tilde{\boldsymbol{x}}_0 + \sum_{i=1}^{t} \sqrt{\alpha_t}\left(\sqrt{\frac{1}{\alpha_i} - 1} - \sqrt{\frac{1}{\alpha_{i-1}} - 1}\right)\epsilon_\theta(\tilde{\boldsymbol{x}}_i, i). \tag{22}$$

Let $\epsilon_\theta(\tilde{\boldsymbol{x}}_i, i) - \epsilon_\theta(\tilde{\boldsymbol{x}}_{i-1}, t) = \tilde{\delta}_i$. Based on Eq. (3), we have:

$$\epsilon_\theta(\tilde{\boldsymbol{x}}_t, t) = -\sqrt{1 - \bar{\alpha}_t} \nabla_{\tilde{\boldsymbol{x}}_{t-1}} \log p_\theta(\tilde{\boldsymbol{x}}_{t-1}) + \tilde{\delta}_t.$$

Therefore, we obtain the following equation for $\tilde{\boldsymbol{x}}_t$:

$$\tilde{\boldsymbol{x}}_t = \sqrt{\alpha_t} \tilde{\boldsymbol{x}}_0 - \sum_{i=0}^{t-1} \nabla_{\tilde{\boldsymbol{x}}_i} \log p_\theta(\tilde{\boldsymbol{x}}_i)^{s_i} + \sum_{i=0}^{t-1} \frac{s_i}{\sqrt{1 - \bar{\alpha}_{i+1}}} \tilde{\delta}_{i+1}. \tag{23}$$

In the initial reversion step, where $\tilde{\boldsymbol{x}}_t = \boldsymbol{x}_t$, we have:

$$\nabla \log p_\theta(\tilde{\boldsymbol{x}}_t) = \nabla \log p_\theta(\boldsymbol{x}_t).$$

Since

$$\nabla \log p_\theta(\tilde{\boldsymbol{x}}_t) = \nabla \log p_\theta(\tilde{\boldsymbol{x}}_{t-1}) - \frac{1}{\sqrt{1 - \bar{\alpha}_t}} \tilde{\delta}_t, \quad \text{and} \quad \nabla \log p_\theta(\boldsymbol{x}_t) = \nabla \log p_\theta(\boldsymbol{x}_{t-1}) - \frac{1}{\sqrt{1 - \bar{\alpha}_t}} \delta_t,$$

we obtain:

$$\nabla \log p_\theta(\tilde{\boldsymbol{x}}_{t-1}) = \nabla \log p_\theta(\boldsymbol{x}_{t-1}) - \frac{1}{\sqrt{1 - \bar{\alpha}_t}} \delta_t + \frac{1}{\sqrt{1 - \bar{\alpha}_t}} \tilde{\delta}_t.$$

By induction, we get:

$$\nabla \log p_\theta(\tilde{\boldsymbol{x}}_{t-k}) = \nabla \log p_\theta(\boldsymbol{x}_{t-k}) + \sum_{j=t-k}^{t-1} \frac{1}{\sqrt{1 - \bar{\alpha}_{j+1}}} \left[ \tilde{\delta}_{j+1} - \delta_{j+1} \right],$$

or more generally:

$$\nabla \log p_\theta(\tilde{\boldsymbol{x}}_i) = \nabla \log p_\theta(\boldsymbol{x}_i) + \sum_{j=i}^{t-1} \frac{1}{\sqrt{1 - \bar{\alpha}_{j+1}}} \left[ \tilde{\delta}_{j+1} - \delta_{j+1} \right].$$

Substituting into Eq. (23), we get:

$$\tilde{\boldsymbol{x}}_t = \sqrt{\alpha_t} \tilde{\boldsymbol{x}}_0 - \sum_{i=0}^{t-1} \nabla_{\boldsymbol{x}_i} \log p_\theta(\boldsymbol{x}_i)^{s_i} - \sum_{i=0}^{t-1} \sum_{j=i}^{t-1} \frac{s_i}{\sqrt{1 - \bar{\alpha}_{j+1}}} \left[ \tilde{\delta}_{j+1} - \delta_{j+1} \right] + \sum_{i=0}^{t-1} \frac{s_i}{\sqrt{1 - \bar{\alpha}_{i+1}}} \tilde{\delta}_{i+1}. \tag{24}$$

Meanwhile, the corresponding equation from Theorem 3.1 is:

$$\boldsymbol{x}_t = \sqrt{\alpha_t} \boldsymbol{x}_0 - \sum_{i=0}^{t-1} [\nabla_{\boldsymbol{x}_i} \log p_\theta(\boldsymbol{x}_i)^{s_i} + \nabla_{\boldsymbol{x}_i} \log p_\theta(y|\boldsymbol{x}_i)^{s_i}] + \sum_{i=0}^{t-1} \frac{s_i}{\sqrt{1 - \bar{\alpha}_{i+1}}} \delta_{i+1}. \tag{25}$$

Equating both sides, we obtain:

$$\sqrt{\alpha_t} \boldsymbol{x}_0 - \sum_{i=0}^{t-1} [\nabla_{\boldsymbol{x}_i} \log p_\theta(\boldsymbol{x}_i)^{s_i} + \nabla_{\boldsymbol{x}_i} \log p_\theta(y|\boldsymbol{x}_i)^{s_i}] + \sum_{i=0}^{t-1} \frac{s_i}{\sqrt{1 - \bar{\alpha}_{i+1}}} \delta_{i+1}$$

$$= \sqrt{\alpha_t} \tilde{\boldsymbol{x}}_0 - \sum_{i=0}^{t-1} \nabla_{\boldsymbol{x}_i} \log p_\theta(\boldsymbol{x}_i)^{s_i} - \sum_{i=0}^{t-1} \sum_{j=i}^{t-1} \frac{s_i}{\sqrt{1 - \bar{\alpha}_{j+1}}} \left[ \tilde{\delta}_{j+1} - \delta_{j+1} \right] + \sum_{i=0}^{t-1} \frac{s_i}{\sqrt{1 - \bar{\alpha}_{i+1}}} \tilde{\delta}_{i+1}. \tag{26}$$

Solving for $\tilde{\boldsymbol{x}}_0$, we arrive at:

$$\tilde{\boldsymbol{x}}_0 = \boldsymbol{x}_0 - \frac{1}{\sqrt{\alpha_t}} \sum_{i=0}^{t-1} \nabla_{\boldsymbol{x}_i} \log p_\theta(y|\boldsymbol{x}_i)^{s_i} + \sum_{i=1}^{t-1} \sum_{j=i}^{t-1} \frac{s_i}{\sqrt{\alpha_t(1 - \bar{\alpha}_{j+1})}} \left[ \tilde{\delta}_{j+1} - \delta_{j+1} \right]. \tag{27}$$

The proof is complete. $\qquad\square$

# B. Additional Experimental Results

## B.1. Experimental Results on 0% Mismatch Proportion across Different Datasets

Tab. 7 presents the results of our proposed method and the compared methods with a 0% mismatch proportion on CIFAR-10, CIFAR-100, and Tiny-ImageNet.

For the open-set task, we observe that UCMD achieves the highest balance score, demonstrating excellent performance even with a 0% mismatch proportion, where no instances from unknown categories are present. This further highlights the effectiveness of the techniques used for generating negative instances. Additionally, we find that MCTF and IOMatch exhibit the same balance score on CIFAR-10, but the high standard deviation in IOMatch results in its lower performance, while MCTF's low mean accuracy contributes to the balance score.

For the closed-set task, we find that UCMD performs slightly below the best accuracy under 0% mismatch proportion. This is because, compared to other mismatch proportions, the number of training instances is smallest in the 0% mismatch scenario, leading to a reduced count of generated positive instances (one realistic instance generates one positive instance for each class). Performance could be further improved by generating more positive instances.

*Table 7.* The average accuracy of methods on the known class (kno.) for the closed-set task, and the balance score (bala.), as well as the accuracy of known (kno.), unknown (unko.), and new classes for the open-set task across the CIFAR-10, CIFAR-100, and Tiny-ImageNet datasets, with a 0% mismatch proportion. The best and second-best results are highlighted in **bold** and underlined, respectively.

| method | CIFAR10 | | | | | CIFAR100 | | | | | Tiny-ImageNet | | | | |
| | closed-set | open-set | | | | closed-set | open-set | | | | closed-set | open-set | | | |
| | kno. | kno. | unkno. | new | bala. | kno. | kno. | unkno. | new | bala. | kno. | kno. | unkno. | new | bala. |
|---|---|---|---|---|---|---|---|---|---|---|---|---|---|---|---|
| DS³L | 70.3 | 70.3 | 0.0 | 0.0 | -17.2 | 21.1 | 21.1 | 0.0 | 0.0 | -5.2 | 25.4 | 25.4 | 0.0 | 0.0 | -6.0 |
| UASD | 78.0 | 78.0 | 0.0 | 0.0 | -19.0 | 26.8 | 26.8 | 0.0 | 0.0 | -6.6 | 5.1 | 5.4 | 0.0 | 0.0 | -1.3 |
| CCSSL | **98.1** | **98.1** | 0.0 | 0.0 | -24.0 | 50.4 | 50.4 | 0.0 | 0.0 | -12.3 | 24.3 | 24.3 | 0.0 | 0.0 | -5.9 |
| T2T | - | - | - | - | - | 54.0 | **54.0** | 0.0 | 0.0 | -13.2 | **40.6** | **40.6** | 0.0 | 0.0 | -9.9 |
| MCTF | 51.9 | 51.3 | 0.0 | 0.0 | -12.5 | **60.1** | 0.0 | **100.0** | **100.0** | 8.9 | 31.7 | 0.0 | 0.0 | 100.0 | -24.4 |
| IOMatch | 96.7 | 96.0 | 12.7 | 5.1 | -12.5 | 30.7 | 0.0 | 100.0 | 100.0 | 8.9 | 32.4 | 0.0 | **100.0** | **100.0** | 8.9 |
| OpenMatch | 96.3 | 95.5 | 7.5 | 3.6 | -16.4 | 7.1 | 6.8 | 14.5 | 12.4 | 7.3 | 25.6 | 24.9 | 20.0 | 19.5 | 18.4 |
| Ours | 94.2 | 91.0 | **100.0** | **96.7** | **91.3** | 46.7 | **33.3** | 92.6 | 92.2 | **38.5** | 32.2 | 16.3 | 91.4 | 89.9 | **22.9** |

## B.2. Experimental Results on Categories with Varying Proportions

To assess the impact of the proportions of known, unknown, and new classes, we conduct experiments by varying the number of these categories while keeping the total instance count fixed. The results are presented in Tab. 8, Tab. 9, and Tab. 10, respectively.

**Impact of known classes.** We investigate the influence of the number of known classes by setting it to 2, 4, and 6, respectively, as presented in Tab. 8.

The experimental results lead to three key observations: *(i)* As the number of known classes increases, the performance of all methods declines, primarily due to the increased complexity of the classification task. *(ii)* For the closed-set task, our method shows competitive performance, achieving results comparable to the best-performing methods across various numbers of known classes. *(iii)* For the open-set task, UCDM achieves the highest balance score across all settings and outperforms all other methods in terms of accuracy for classifying both known and new classes.

These findings underscore the robustness and adaptability of the proposed method in handling varying numbers of known classes across both closed-set and open-set tasks.

**Impact of unknown classes.** We investigate the effect of varying the number of unknown classes by setting it to 2, 4, and 6, as detailed in Tab. 9.

The experimental results lead to two important observations: *(i)* UCDM demonstrates robustness to varying numbers of unknown categories, primarily due to its negative instance generation pipeline. By erasing semantic class information from images and generating instances that closely resemble the original ones, this approach ensures effective handling of the challenges posed by different proportions of unknown categories. *(ii)* UCMD achieves the second-highest accuracy for the

*Table 8.* The average accuracy of methods on the known class (kno.) for the closed-set task, the balance score (bala.), and the accuracy of known (kno.), unknown (unko.), and new classes for the open-set task on the CIFAR-10 dataset, with varying proportions of known classes. The item "4/2/2" denotes four known classes, two unknown classes, and two new classes. The best and second-best results are highlighted in **bold** and underlined, respectively.

| method | 2/2/2 closed-set kno. | open-set kno. | unkno. | new | bala. | 4/2/2 closed-set kno. | open-set kno. | unkno. | new | bala. | 6/2/2 closed-set kno. | open-set kno. | unkno. | new | bala. |
|---|---|---|---|---|---|---|---|---|---|---|---|---|---|---|---|
| DS³L | 72.7 | 72.7 | 0.0 | 0.0 | -17.7 | 51.4 | 51.4 | 0.0 | 0.0 | -12.5 | 42.1 | 42.1 | 0.0 | 0.0 | -10.3 |
| UASD | 79.2 | 79.2 | 0.0 | 0.0 | -19.3 | 56.4 | 56.4 | 0.0 | 0.0 | -13.8 | 42.8 | 42.8 | 0.0 | 0.0 | -10.5 |
| CCSSL | **96.4** | **96.4** | 0.0 | 0.0 | -23.5 | 79.1 | 79.1 | 0.0 | 0.0 | -19.3 | 65.5 | 65.5 | 0.0 | 0.0 | -16.0 |
| T2T | - | - | - | - | - | **81.4** | **81.4** | 0.0 | 0.0 | -19.9 | **69.6** | **69.6** | 0.0 | 0.0 | -17.0 |
| MCTF | 62.7 | 62.7 | 0.0 | 0.0 | -15.3 | 61.8 | 61.9 | 0.0 | 0.0 | -15.1 | 52.1 | 51.9 | 0.0 | 0.0 | -12.7 |
| OpenMatch | 70.2 | 61.1 | 49.8 | 37.9 | 38.0 | 68.8 | 60.1 | 15.6 | 34.0 | 14.2 | 35.9 | 24.9 | 76.7 | 55.4 | 26.3 |
| IOMatch | 90.0 | 87.8 | 7.1 | 6.1 | -13.2 | 72.0 | 66.4 | 12.6 | 17.4 | 2.3 | 56.4 | 26.7 | 65.4 | 54.4 | 28.9 |
| Ours | 93.2 | 91.9 | **100.0** | **98.1** | **92.5** | 75.9 | 69.1 | **95.2** | **98.9** | **71.5** | 64.3 | 56.7 | **96.5** | **94.4** | **60.2** |

*Table 9.* The average accuracy of methods on the known class (kno.) for the closed-set task, the balance score (bala.), and the accuracy of known (kno.), unknown (unko.), and new classes for the open-set task on the CIFAR-10 dataset, with varying proportions of unknown classes. The item "2/4/2" indicates two known, four unknown, and two new classes. The best and second-best results are highlighted in **bold** and underlined, respectively.

| method | 2/2/2 closed-set kno. | open-set kno. | unkno. | new | bala. | 2/4/2 closed-set kno. | open-set kno. | unkno. | new | bala. | 2/6/2 closed-set kno. | open-set kno. | unkno. | new | bala. |
|---|---|---|---|---|---|---|---|---|---|---|---|---|---|---|---|
| DS³L | 72.7 | 72.7 | 0.0 | 0.0 | -17.7 | 67.1 | 67.1 | 0.0 | 0.0 | -16.4 | 71.8 | 71.8 | 0.0 | 0.0 | -17.5 |
| UASD | 79.2 | 79.2 | 0.0 | 0.0 | -19.3 | 79.0 | 79.0 | 0.0 | 0.0 | -19.3 | 74.0 | 74.0 | 0.0 | 0.0 | -18.1 |
| CCSSL | **96.4** | **96.4** | 0.0 | 0.0 | -23.5 | **97.2** | **97.2** | 0.0 | 0.0 | -23.7 | **97.0** | **97.0** | 0.0 | 0.0 | -23.7 |
| T2T | - | - | - | - | - | - | - | - | - | - | - | - | - | - | - |
| MCTF | 62.7 | 62.7 | 0.0 | 0.0 | -15.3 | 65.8 | 65.8 | 0.0 | 0.0 | -16.0 | 54.3 | 54.3 | 0.0 | 0.0 | -13.3 |
| OpenMatch | 70.2 | 61.1 | 49.8 | 37.9 | 38.0 | 76.8 | 75.9 | 2.6 | 3.1 | -15.1 | 75.7 | 73.5 | 15.0 | 11.9 | -1.3 |
| IOMatch | 90.0 | 87.8 | 7.1 | 6.1 | -13.2 | 89.4 | 87.1 | 5.1 | 6.3 | -14.2 | 89.2 | 87.3 | 6.2 | 5.8 | -13.8 |
| Ours | 93.2 | 91.9 | **100.0** | **98.1** | **92.5** | 94.5 | 89.1 | **100.0** | **97.5** | **89.8** | 95.9 | 94.9 | **100.0** | **99.2** | **95.3** |

*Table 10.* The average accuracy of methods on the known class (kno.) for the closed-set task, the balance score (bala.), and the accuracy of known (kno.), unknown (unko.), and new classes for the open-set task on the CIFAR-10 dataset, with varying proportions of new classes. The item "2/2/4" denotes two known, two unknown, and four new classes. The best and second-best results are highlighted in **bold** and underlined, respectively.

| method | 2/2/2 closed-set kno. | open-set kno. | unkno. | new | bala. | 2/2/4 closed-set kno. | open-set kno. | unkno. | new | bala. | 2/2/6 closed-set kno. | open-set kno. | unkno. | new | bala. |
|---|---|---|---|---|---|---|---|---|---|---|---|---|---|---|---|
| DS³L | 72.7 | 72.7 | 0.0 | 0.0 | -17.7 | 72.7 | 72.7 | 0.0 | 0.0 | -17.7 | 72.7 | 72.7 | 0.0 | 0.0 | -17.7 |
| UASD | 79.2 | 79.2 | 0.0 | 0.0 | -19.3 | 79.2 | 79.2 | 0.0 | 0.0 | -19.3 | 79.2 | 79.2 | 0.0 | 0.0 | -19.3 |
| CCSSL | **96.4** | **96.4** | 0.0 | 0.0 | -23.5 | **96.4** | **96.4** | 0.0 | 0.0 | -23.5 | **96.4** | **96.4** | 0.0 | 0.0 | -23.5 |
| T2T | - | - | - | - | - | - | - | - | - | - | - | - | - | - | - |
| MCTF | 62.7 | 62.7 | 0.0 | 0.0 | -15.3 | 62.7 | 62.7 | 0.0 | 0.0 | -15.3 | 62.7 | 62.7 | 0.0 | 0.0 | -15.3 |
| OpenMatch | 70.2 | 61.1 | 49.8 | 37.9 | 38.0 | 70.2 | 61.1 | 49.8 | 42.2 | 41.5 | 70.2 | 61.1 | 49.8 | 43.1 | 42.2 |
| IOMatch | 90.0 | 87.8 | 7.1 | 6.1 | -13.2 | 90.0 | 87.8 | 7.1 | 7.9 | -12.1 | 90.0 | 87.8 | 7.1 | 7.8 | -12.1 |
| Ours | 93.2 | 91.9 | **100.0** | **98.1** | **92.5** | 93.2 | 91.9 | **100.0** | **100.0** | **92.6** | 93.2 | 91.9 | **100.0** | **100.0** | **92.6** |

closed-set task and the highest balance score for the open-set task across all settings. This suggests that UCMD remains unaffected by variations in the count of unknown categories.

**Impact of new classes.** We explore the impact of varying the number of new classes by setting it to 2, 4, and 6, as shown in Tab. 10.

From the results, we observe that UCMD demonstrates strong generalization performance, remaining insensitive to the count of new classes, and achieves the best balance score. This indicates that the proposed method is robust to changes in the number of new classes and can effectively handle open-set classification tasks.

### B.3. Experimental Results on Generated Positive Instances with Varying Parameter $\sigma_t$

We evaluate the impact of random noise strength $\sigma_t$ in the positive instance generation pipeline by setting it to 0 and 1 (our setting), as shown in Fig. 7.

The results show that setting $\sigma_t = 1$ yields the best performance in both known-class accuracy on the closed-set task and balance score on the open-set task. This indicates that increasing the strength of random noise in the positive instance generation pipeline introduces more diversity into the training process.

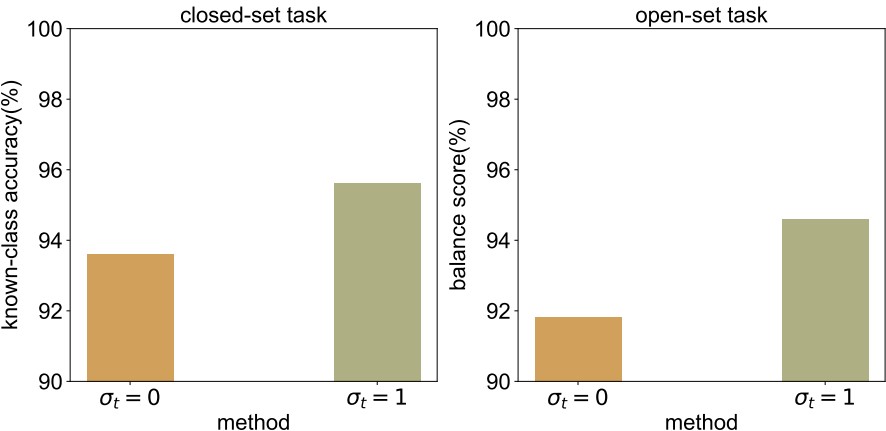

*Figure 7.* Experimental results comparing the generated positive instances with random noise strengths $\sigma_t = 0$ and $\sigma_t = 1$ (our setting), respectively, on CIFAR-10 with a 60% mismatch proportion.

### B.4. Experimental Results on Generated Negative Instances with Varying Parameter $\sigma_t$

We evaluate the impact of random noise strength $\sigma_t$ in the negative instance generation pipeline by setting it to 1 and 0.2 (our setting), as shown in Fig. 8.

The results show that setting $\sigma_t = 0.2$ yields the best performance, especially in terms of the balance score. This suggests that using a smaller $\sigma_t$ in the negative instance generation pipeline helps achieve a more effective contrast with positive instances.

### B.5. Analysis of the Sensitivity to Weights in the Loss Function

We investigate the impact of the parameters $\lambda_1$ and $\lambda_2$, which balance the weights of detection and classification tasks, on CIFAR-10 with a 60% mismatch proportion, as illustrated in Fig. 9. To reflect overall performance in the open-set task, we report the balance score.

The results reveal the following findings: *(i)* The solid-line trend remains stable across all tasks with varying $\lambda_1$, indicating that performance is largely insensitive to this parameter. However, values between 1 and 3 tend to yield better results. *(ii)* When $\lambda_2$ is set between 2 and 5, the results consistently surpass those achieved with $\lambda_2 = 1$, particularly in the closed-set task. This suggests that tuning $\lambda_2$ based on the classification task's complexity can improve performance. *(iii)* The trends observed in the closed-set task align closely with those in the open task, highlighting the interdependence between detection

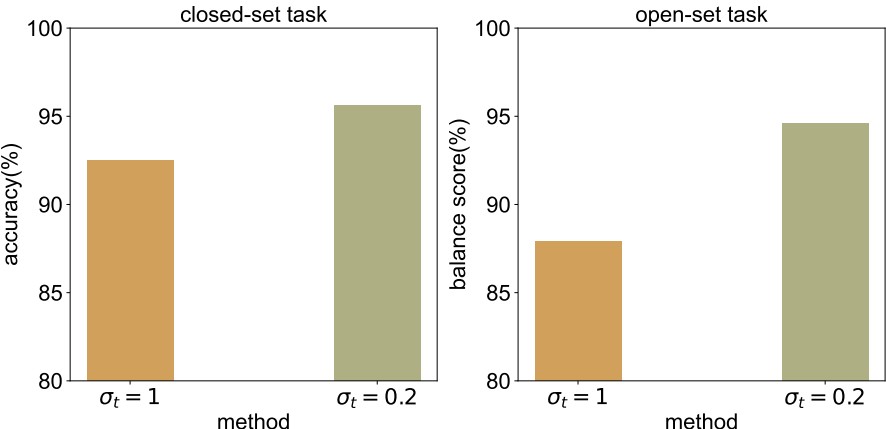

*Figure 8.* Experimental results comparing the generated negative instances with random noise strengths $\sigma_t = 1$ and $\sigma_t = 0.2$ (our setting), respectively, on CIFAR-10 with a 60% mismatch proportion.

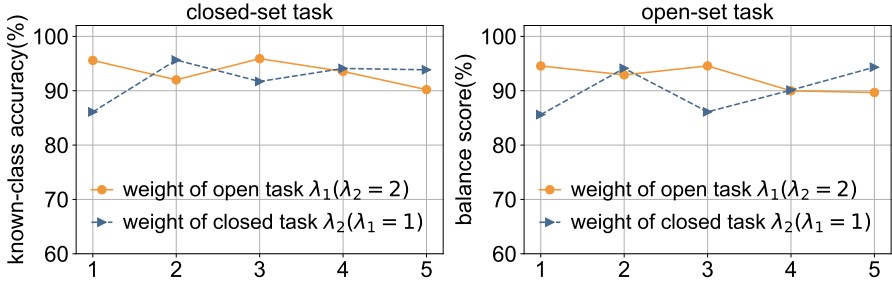

*Figure 9.* Loss weight configurations.

and classification tasks.

## B.6. Evaluation of the Impact of the Batch Normalization Layer on Model Training

Several studies (Oliver et al., 2018; Zhao et al., 2020; 2021; 2022b;a; 2024c;a) demonstrate the significant impact of noisy data on models with batch normalization (BN). Noisy data affects the estimation of the mean and variance during the BN process, leading to poor BN representations and preventing the model from learning optimal BN parameters ($\gamma_{bn}$ and $\beta_{bn}$). Therefore, we suggest updating the parameters $\gamma_{bn}$ and $\beta_{bn}$ when training solely with generated positive and negative instances. This approach helps mitigate the negative impact of instances with incorrect pseudo-labels and enables the model to learn better BN parameters.

Tab. 10 compares our method with a variant approach of updating BN during training for the closed-set and open-set tasks on CIFAR-10 with a 60% mismatch proportion. Clearly, our method outperforms the latter on both tasks. This demonstrates that training with generated positive and negative instances contributes to learning better BN representations and facilitates more effective model training.

## B.7. Evaluation of Pseudo-Label Reliability in Selected Instances

To evaluate the reliability of pseudo-labels assigned to the selected instances in the confidence-based labeling mechanism, we present the count of instances with accurate pseudo-labels and the count of selected instances on CIFAR-10 under 0% and 75% mismatch proportions in Fig. 11

From the results, we have the following two findings.

*(i)* The proportion of instances with accurate pseudo-labels relative to the selected instances is consistently high under both

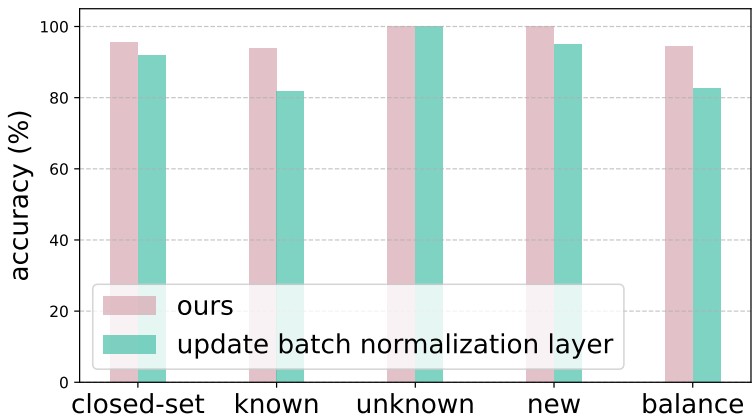

*Figure 10.* Ablation study of batch normalization layer.

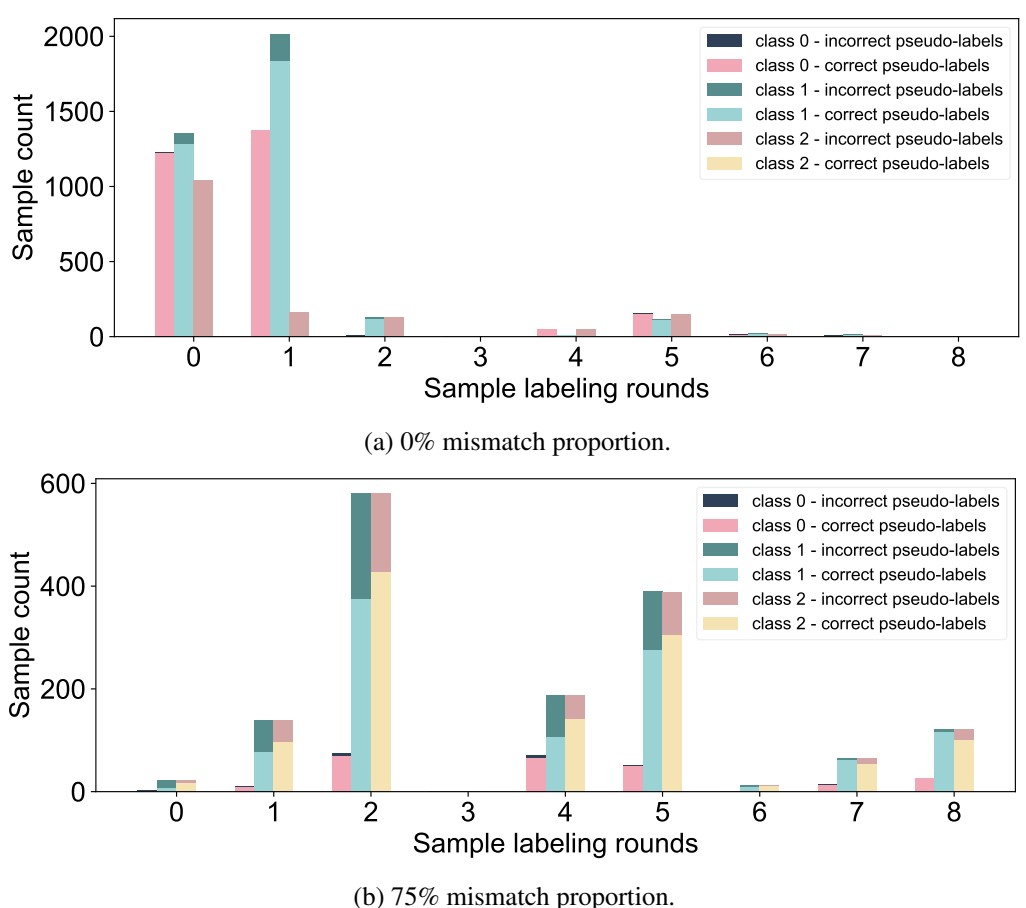

(a) 0% mismatch proportion.

(b) 75% mismatch proportion.

*Figure 11.* Counts of selected samples and correctly pseudo-labeled samples across categories for each selection round, with class 2 representing the unified other category in testing, including both unknown classes from the training data and new classes introduced during testing.

0% and 75% mismatch proportions, highlighting the effectiveness of the confidence-based labeling. Notably, under a 0% mismatch proportion, no instances in class 2 have accurate pseudo-labels, as all instances belong to the known classes.

*(ii)* As the selection rounds progress, the number of selected instances decreases, while the proportion of instances with accurate pseudo-labels among the selected instances increases. This indicates that the model becomes more stable over successive rounds.

### B.8. Visualization of Generated Positive Images from Random Noise

We compare the positive instances generated by UCDM, starting from the seed sample, with those produced using conditional guidance starting from random noise, as shown in Fig. 12.

Notably, the positive instances generated by UCDM not only exhibit the semantic class specified in the prompt but also preserve the style of the seed sample, effectively mitigating the potential negative impact of domain shift.

Meanwhile, the "papillon" images correspond to a dog in the original image, while the randomly generated images depict a butterfly. This discrepancy arises from polysemy—multiple semantic meanings or physical instantiations of class names used as prompts. However, our method starts from the latent of the original image, exhibiting the expected semantics.

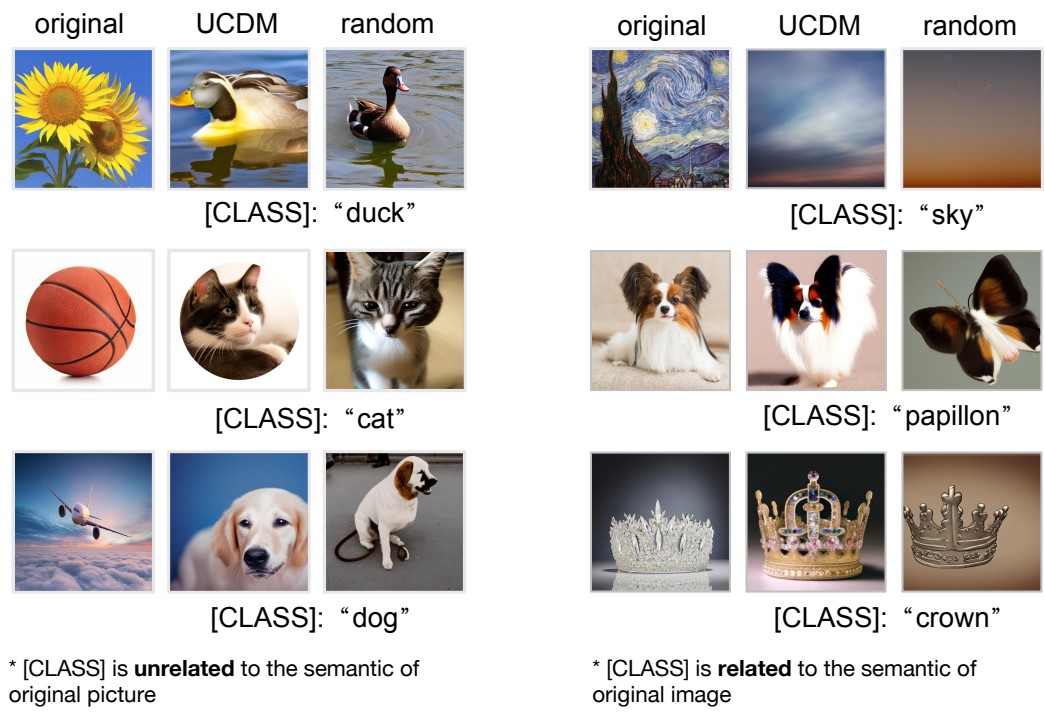

*Figure 12.* Visualization of positive instances generated by UCDM and random noise, with the prompt "A photo of a [CLASS]". The specific "[CLASS]" is indicated below each image.

### B.9. Visualization of Generated Negative Images from Random Noise

We compare the negative instances generated by UCDM with those produced using unconditional guidance starting from random noise, as shown in Fig. 13.

It is evident that when the semantic class in the images is unrelated to the prompt, UCDM tends to preserve the semantics of the original image. In contrast, when the semantic class in the image matches the prompt, UCDM effectively erases the semantic class. However, the randomly generated images fail to achieve this behavior.

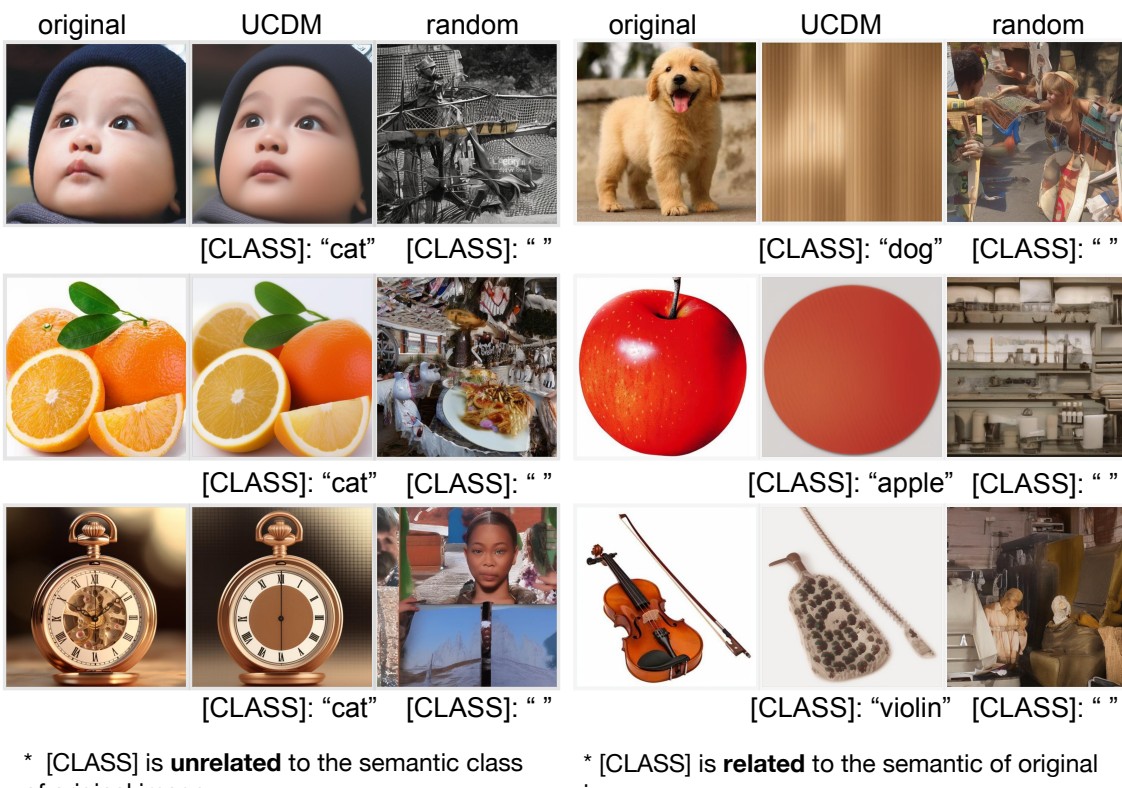

*Figure 13.* Visualization of negative instances generated by UCDM, with the prompt "A photo of a [CLASS]", and random noise. The specific "[CLASS]" is indicated below each image.

## B.10. Visualization of Generated Positive Images with Varying Parameter $\sigma_t$

We present visualizations of positive instances generated by UCDM under varying levels of random noise strength ($\sigma_t$), with the prompt set to "A photo of a [CLASS]", as shown in Fig. 14.

From the results, we observe that as the random noise strength $\sigma_t$ increases, the generated images exhibit greater diversity while preserving the style and key visual characteristics, such as structure and color, of the original ones.

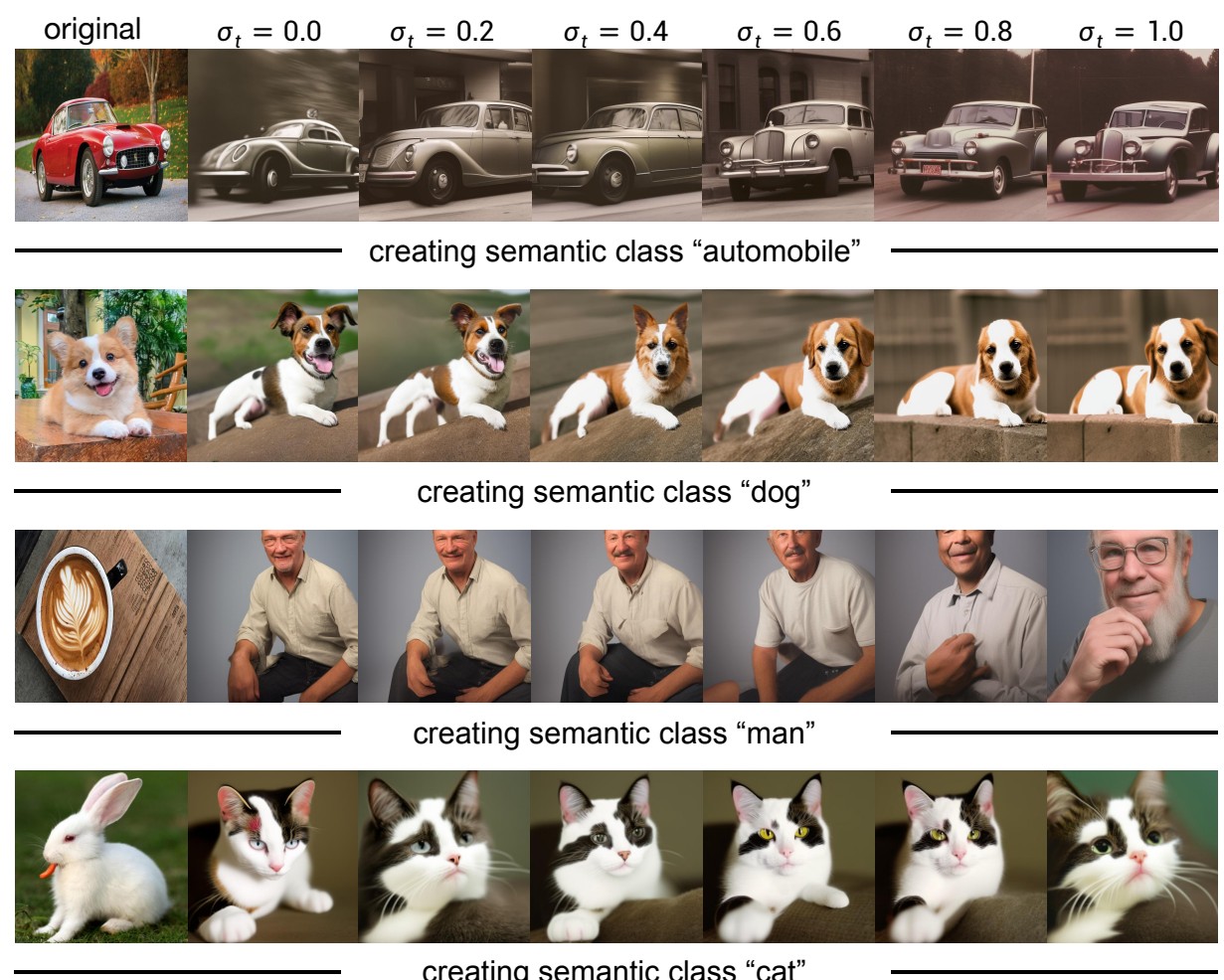

*Figure 14.* Visualization of positive instances generated by UCDM under varying random noise strengths ($\sigma_t$), with the prompt "A photo of a [CLASS]". Here, "[CLASS]" is specified as "automobile", "dog", "man", and "cat".

## B.11. Visualization of Generated Negative Images with Varying Parameter $\sigma_t$

We visualize the negative instances generated by UCDM under varying levels of random noise strength ($\sigma_t$), with the prompt set to "A photo of a dog", in Fig. 15.

The results indicate that as the random noise strength $\sigma_t$ increases, the discrepancy between the generated image and the original one also grows. For images that do not match the semantic class, the original image may become distorted when $\sigma_t \geq 0.4$. Conversely, for images that do match the semantic class, the visual fidelity of the generated images improves.

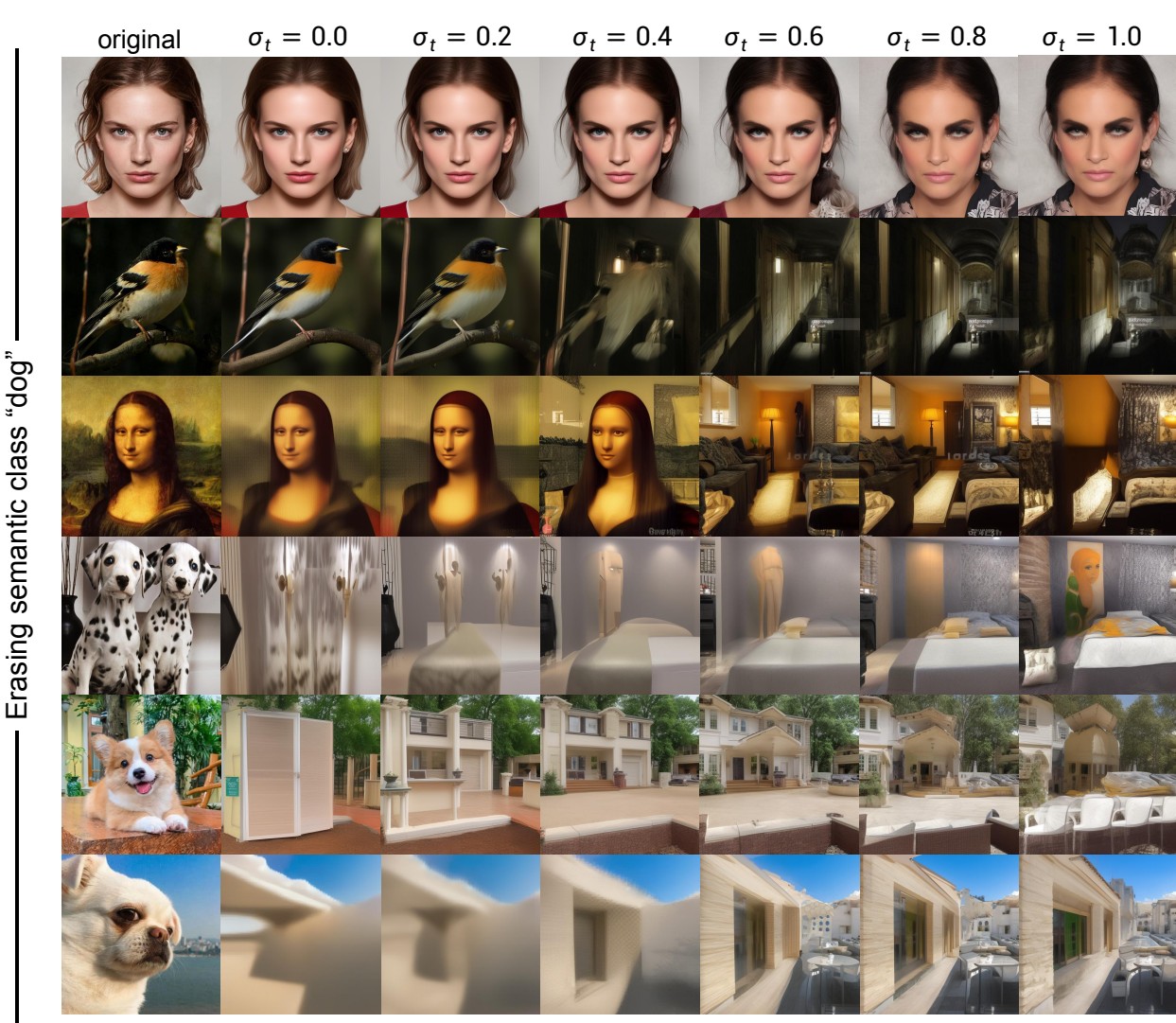

*Figure 15.* Visualization of negative instances generated by UCDM under varying random noise strengths ($\sigma_t$), with the prompt set to "A photo of a dog".

## B.12. Visualization of DDIM Inversion and Negative Instances Generation in UCMD

We compare our generated negative instance with instances generated using DDIM inversion and unconditional reverse, as shown in Fig. 16.

As illustrated in Fig. 16, the difference between DDIM inversion and conditional inversion creates a distinct gap between the two generated images. Our negative generation pipeline effectively erases the semantic class, while DDIM inversion preserves the original image. This demonstrates that the semantic class is erased in conditional inversion, as formally stated in Theorem 3.1.

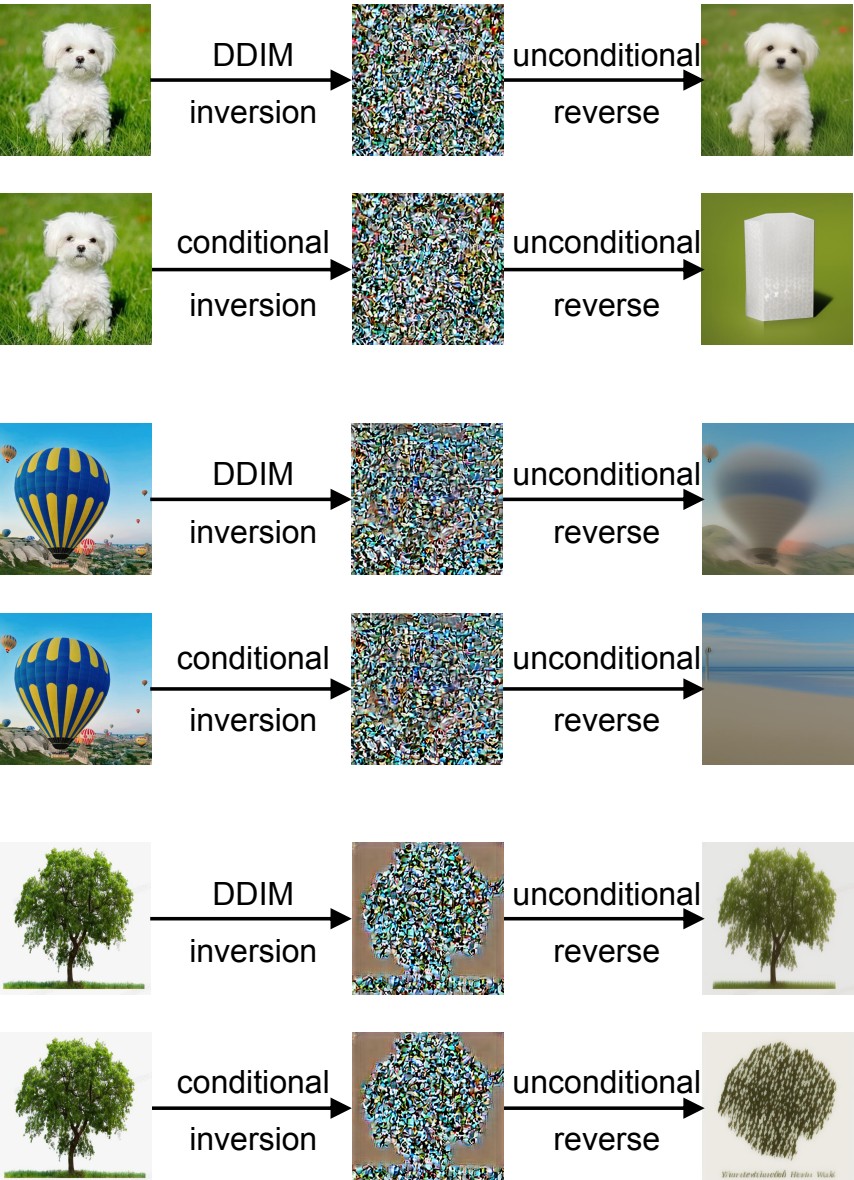

*Figure 16.* Visualization of negative instances generated by UCDM and DDIM inversion, using the prompt "A photo of a [CLASS]." Here, [CLASS] is set to "A photo of a [CLASS]". Here, [CLASS] is set to "dog", "hot air balloon", and "tree", respectively.

## B.13. Visualization of Generated Positive and Negative Images in UCMD

We present visualizations of the positive and negative instances generated by UCDM in Fig. 17, Fig. 18, Fig. 19, and Fig. 20, using seed samples from the ImageNet (Deng et al., 2009) dataset.

The results demonstrate that our negative instance generation pipeline effectively removes the semantic class from the images while preserving the original image characteristics if they do not match the semantic class.

In addition, our positive instance generation pipeline produces instances that retain the style of the original image and accurately reflect the semantic class specified in the prompt. Furthermore, the generated positive instances exhibit diversity.

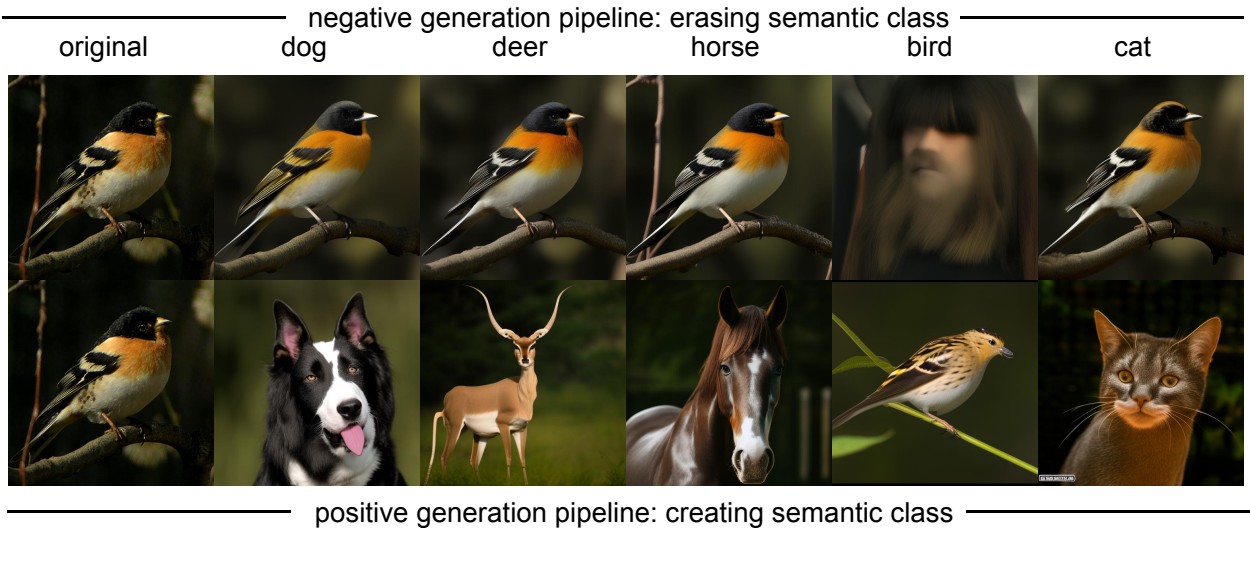

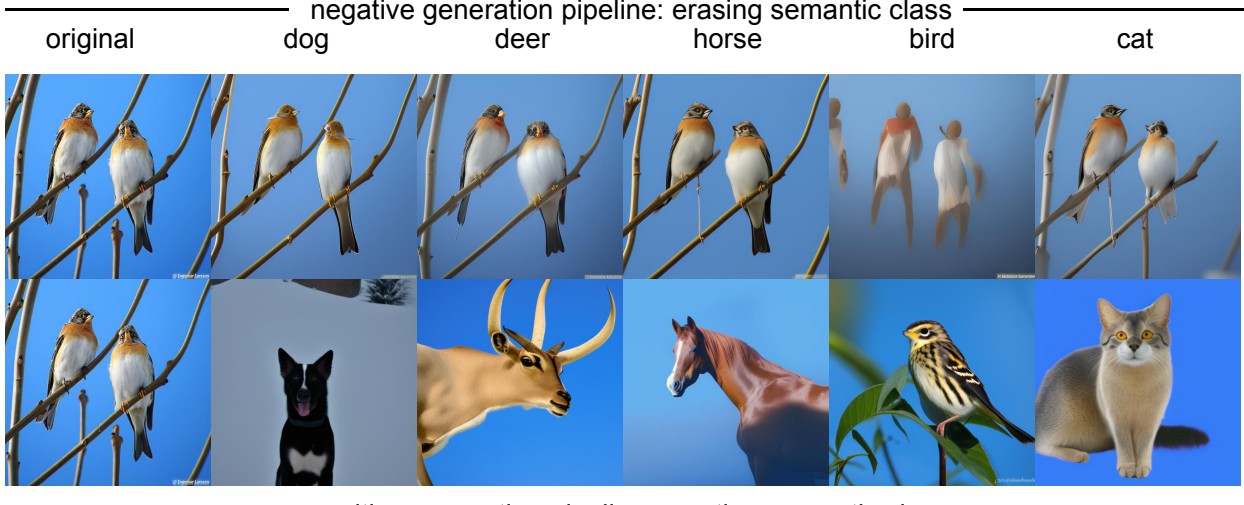

*Figure 17.* Visualization of positive and negative instances generated by UCDM, where the seed sample is a bird, and semantic classes such as "dog", "deer", "horse", "bird", and "cat" are created or erased using the prompt "A photo of a [CLASS]".

negative generation pipeline: erasing semantic class

| original | dog | deer | horse | bird | cat |

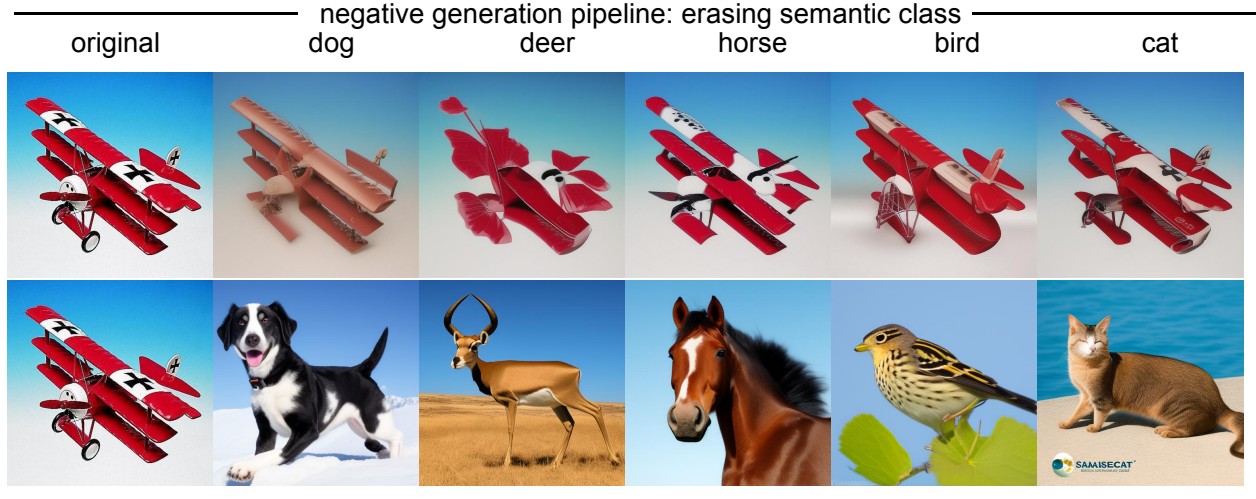

positive generation pipeline: creating semantic class

negative generation pipeline: erasing semantic class

| original | dog | deer | horse | bird | cat |

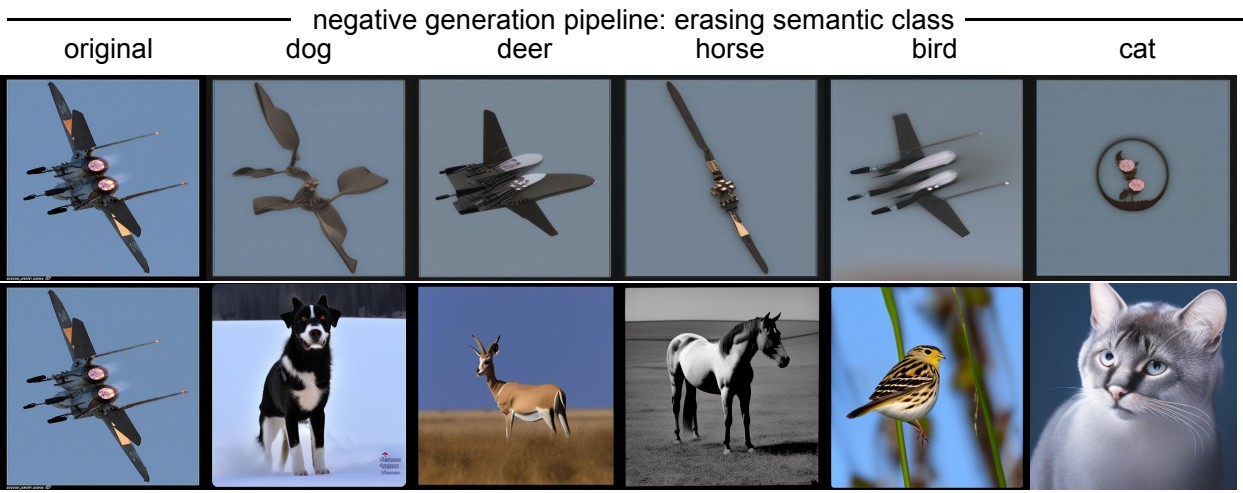

positive generation pipeline: creating semantic class

*Figure 18.* Visualization of positive and negative instances generated by UCDM, where the seed sample is an airplane, and semantic classes such as "dog", "deer", "horse", "bird", and "cat" are created or erased using the prompt "A photo of a [CLASS]".

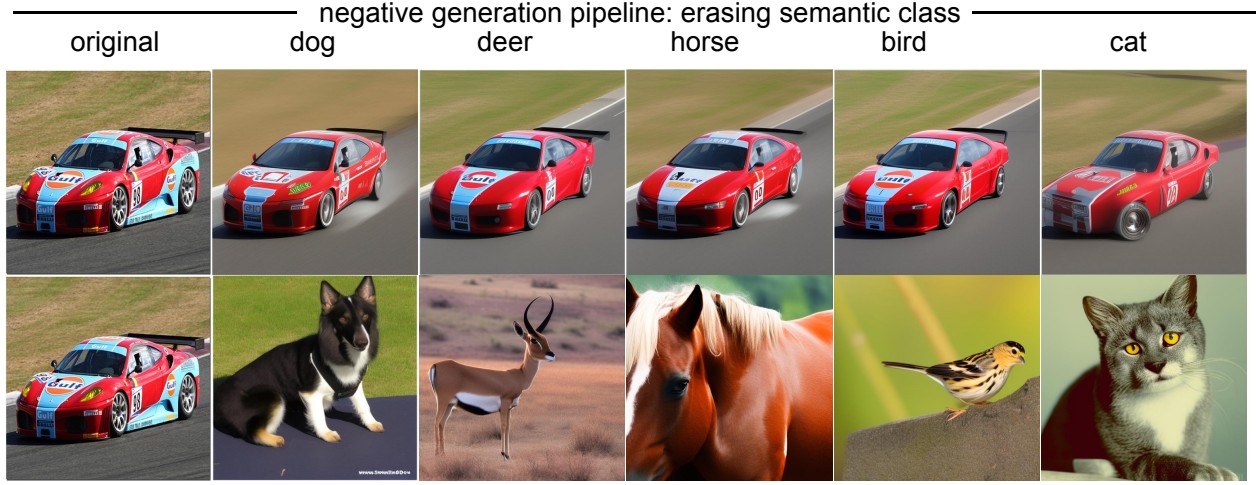

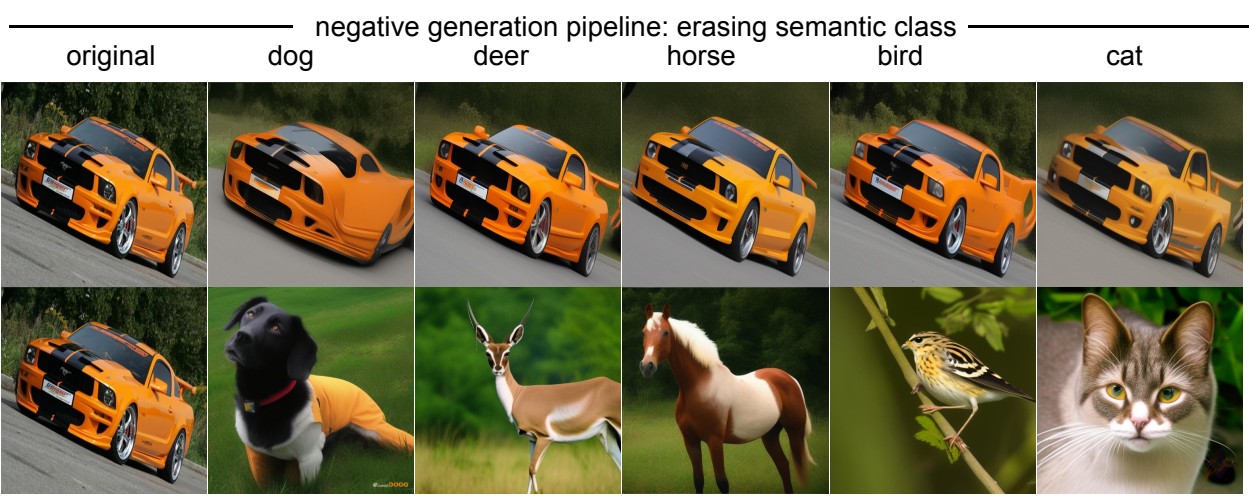

*Figure 19.* Visualization of positive and negative instances generated by UCDM, where the seed sample is a car, and semantic classes such as "dog", "deer", "horse", "bird", and "cat" are created or erased using the prompt "A photo of a [CLASS]".

negative generation pipeline: erasing semantic class

| original | dog | deer | horse | bird | cat |

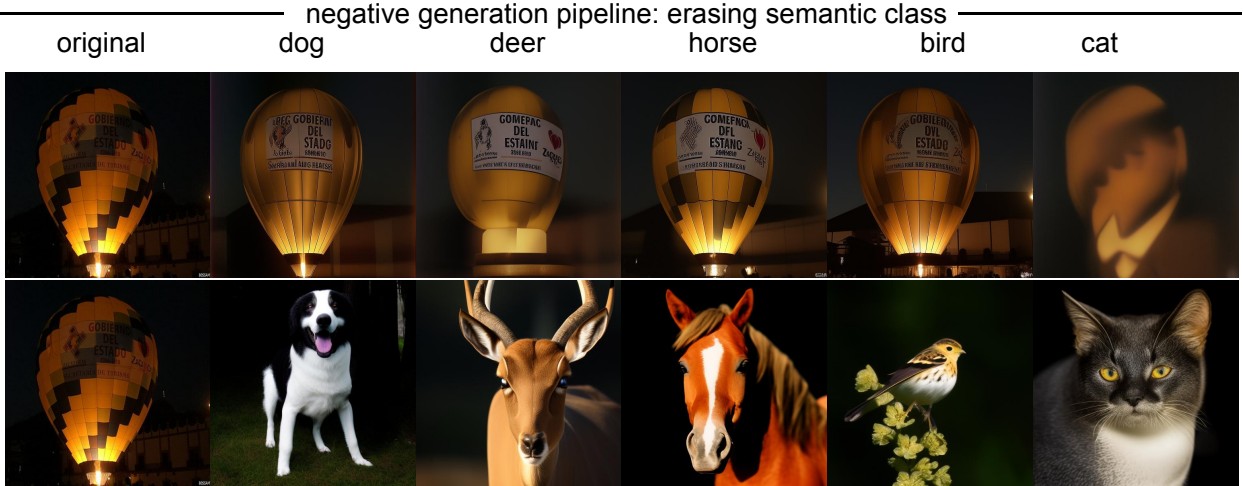

positive generation pipeline: creating semantic class

negative generation pipeline: erasing semantic class

| original | dog | deer | horse | bird | cat |

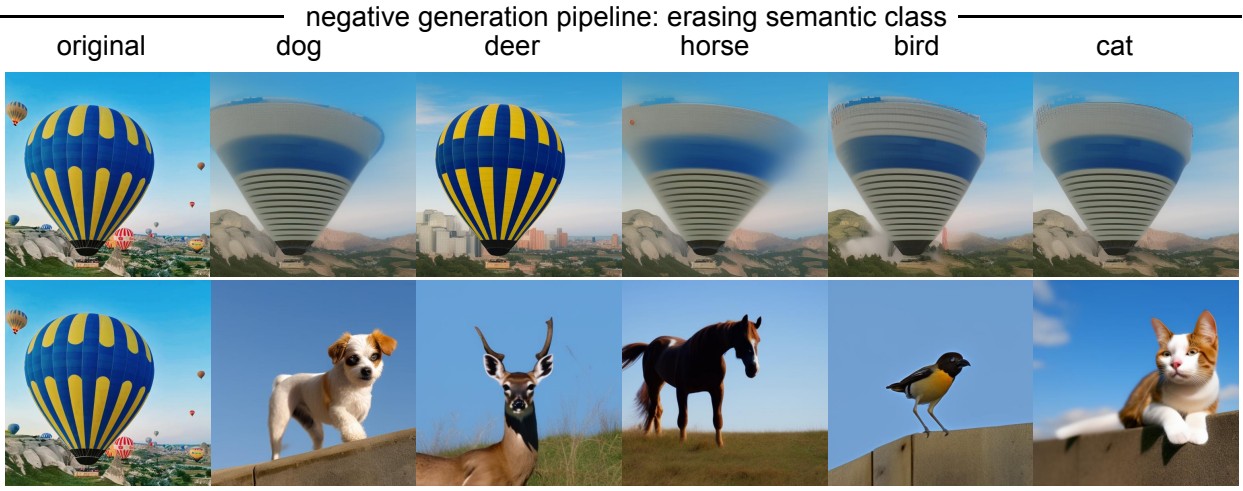

positive generation pipeline: creating semantic class

*Figure 20.* Visualization of positive and negative instances generated by UCDM, where the seed sample is a hot air balloon, and semantic classes such as "dog", "deer", "horse", "bird", and "cat" are created or erased using the prompt "A photo of a [CLASS]".

## B.14. Visualization of generated harder training pairs

By designing more specific prompts to erase critical features, we generate harder training pairs with high visual similarity and improved contrast. As shown in Fig. 21, this highlights the effectiveness of UCDM in producing challenging examples for training.

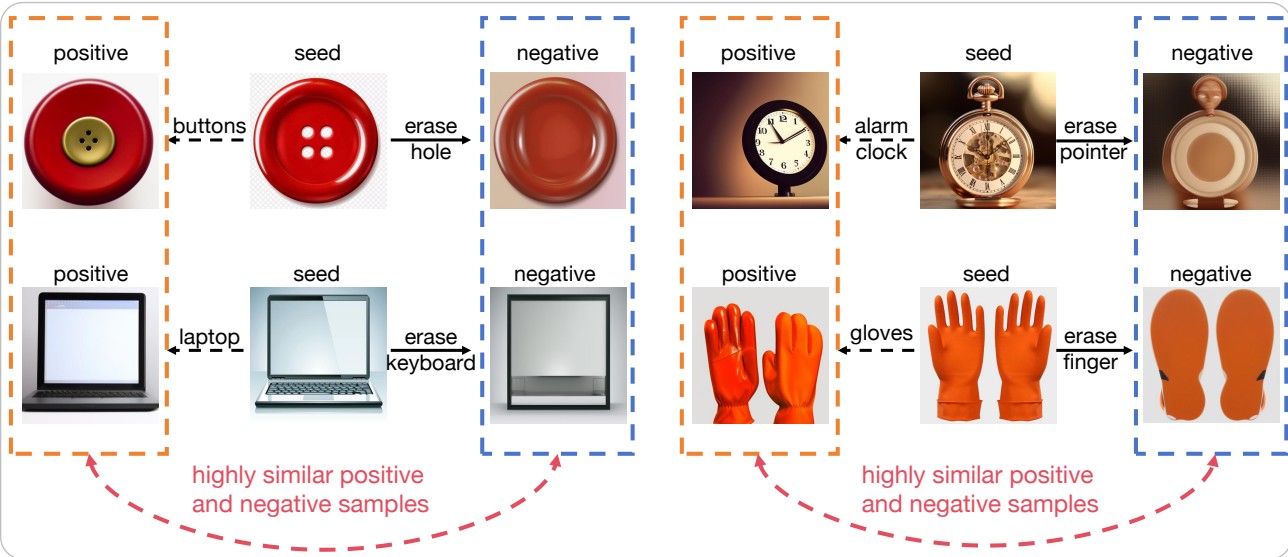

*Figure 21.* Generated hard training pairs where positive and negative instances are highly similar. Negative instances are created by guiding the generation process with a carefully designed prompt, such as "erase hole from button." This approach removes crucial features from seed samples and shifts the original semantics.

# C. Experimental Settings

The section provides a detailed overview of the datasets and training procedures.

## C.1. Dataset Details

This section provides detailed information about the datasets, including CIFAR-10 (Krizhevsky et al., 2009), CIFAR-100 (Krizhevsky et al., 2009), and Tiny-ImageNet (Deng et al., 2009). The CIFAR-10 and CIFAR-100 dataset comprises 50,000 training and 10,000 testing images of 10 and 100 categories, respectively. Tiny-ImageNet is the subset of ImageNet that contains 100,000 training and 10,000 testing images across 200 categories.

The known, unknown, and new classes in CIFAR-10, CIFAR-100, and Tiny-ImageNet are detailed in Tab. 11, Tab. 12, and Tab. 13, respectively. Additionally, Tab. 14 and Tab. 15 present the instance counts for known, unknown, and new classes across the training and testing sets of CIFAR-10, CIFAR-100, and Tiny-ImageNet, respectively.

*Table 11.* The class names of known, unknown, and new classes in CIFAR-10.

| type | class name |
|---|---|
| known class | airplane, automobile |
| unknown class | bird, cat, deer, dog, frog, |
| new class | horse, ship, truck |

*Table 12.* The class names of known, unknown, and new classes in CIFAR-100.

| type | class name |
|---|---|
| known class | bear, camel, cattle, chimpanzee, flatfish, girl, house, keyboard, leopard, lion, mouse, porcupine, possum, rabbit, raccoon, shrew, skunk, squirrel, tiger, wolf, |
| unknown class | apple, aquarium fish, baby, beaver, bed, bee, beetle, bicycle, bottle, bowl, boy, bridge, bus, butterfly, can, castle, caterpillar, chair, clock, cloud, cockroach, couch, crab, crocodile, cup, dinosaur, dolphin, elephant, forest, fox, hamster, kangaroo, lamp, lawn mower, lizard, lobster, man, maple tree, motorcycle, mountain, mushroom, oak tree, orange, orchid, otter, palm tree, pear, pickup truck, pine tree, plain, plate, poppy, ray, road, rocket, rose, sea, seal, shark, skyscraper, |
| new class | snail, snake, spider, streetcar, sunflower, sweet pepper, table, tank, telephone, television, tractor, train, trout, tulip, turtle, wardrobe, whale, willow tree, woman, worm |

*Table 13.* The class names of known, unknown, and new classes in Tiny-ImageNet.

| type | class name |
|---|---|
| known class | goldfish, fire salamander, American bullfrog, tailed frog, American alligator, boa constrictor, trilobite, scorpion, southern black widow, tarantula, centipede, goose, koala, jellyfish, brain coral, snail, slug, sea slug, American lobster, spiny lobster |
| unknown class | black stork, king penguin, albatross, dugong, Chihuahua, Yorkshire Terrier, Golden Retriever, Labrador Retriever, German Shepherd Dog, Standard Poodle, tabby cat, Persian cat, Egyptian Mau, cougar, lion, brown bear, ladybug, fly, bee, grasshopper, stick insect, cockroach, praying mantis, dragonfly, monarch butterfly, sulphur butterfly, sea cucumber, guinea pig, pig, ox, bison, bighorn sheep, gazelle, arabian camel, orangutan, chimpanzee, baboon, African bush elephant, red panda, abacus, academic gown, altar, apron, backpack, baluster / handrail, barbershop, barn, barrel, basketball, bathtub, station wagon, lighthouse, beaker, beer bottle, bikini, binoculars, birdhouse, bow tie, brass memorial plaque, broom, bucket, high-speed train, butcher shop, candle, cannon, cardigan, automated teller machine, CD player, chain, storage chest, Christmas stocking, cliff dwelling, computer keyboard, candy store, convertible, construction crane, dam, desk, dining table, drumstick |
| new class | dumbbell, flagpole, fountain, freight car, frying pan, fur coat, gas mask or respirator, go-kart, gondola, hourglass, iPod, rickshaw, kimono, lampshade, lawn mower, lifeboat, limousine, magnetic, compass, maypole, military uniform, miniskirt, moving van, metal nail, neck brace, obelisk, oboe, pipe organ, parking meter, payphone, picket fence, pill bottle, plunger, pole, police van, poncho, soda bottle, potter's wheel, missile, punching bag, fishing casting reel, refrigerator, remote control, rocking chair, rugby ball, sandal, school bus, scoreboard, sewing machine, snorkel, sock, sombrero, space heater, spider web, sports car, through arch bridge, stopwatch, sunglasses, suspension bridge, swim trunks / shorts, syringe, teapot, teddy bear, thatched roof, torch, tractor, triumphal arch, trolleybus, turnstile, umbrella, vestment, viaduct, volleyball, water jug, water tower, wok, wooden spoon, comic book, plate, guacamole, ice cream, popsicle, pretzel, mashed potatoes, cauliflower, bell pepper, mushroom, orange, lemon, banana, pomegranate, meatloaf, pizza, pot pie, espresso, mountain, cliff, coral reef, lakeshore, beach, acorn |

*Table 14.* The counts of instances for known (kno.) and unknown (unkno.) classes in the training sets of CIFAR-10, CIFAR-100, and Tiny-ImageNet datasets, with mismatch proportions ranging from 0% to 75%.

| dataset | category | | 0% | | 20% | | 40% | | 60% | | 75% | |
|---|---|---|---|---|---|---|---|---|---|---|---|---|
| | kno. | unkno. | kno. | unkno. | kno. | unkno. | kno. | unkno. | kno. | unkno. | kno. | unkno. |
| CIFAR-10 | 2 | 5 | 10,000 | 0 | 10,000 | 2,500 | 10,000 | 6,667 | 10,000 | 15,000 | 10,000 | 3,0000 |
| CIFAR-100 | 20 | 60 | 10,000 | 0 | 10,000 | 2,500 | 10,000 | 6,667 | 10,000 | 15,000 | 10,000 | 3,0000 |
| Tiny-ImageNet | 20 | 80 | 10,000 | 0 | 10,000 | 2,500 | 10,000 | 6,667 | 10,000 | 15,000 | 10,000 | 3,0000 |

*Table 15.* The counts of instances for known (kno.), unknown (unkno.), and new classes in the testing sets of CIFAR-10, CIFAR-100, and Tiny-ImageNet datasets.

| dataset | category | | | count | | |
|---|---|---|---|---|---|---|
| | kno. | unkno. | new | kno. | unkno. | new |
| CIFAR-10 | 2 | 5 | 3 | 2,000 | 2,000 | 2,000 |
| CIFAR-100 | 20 | 60 | 20 | 2,000 | 2,000 | 2,000 |
| Tiny-ImageNet | 20 | 80 | 100 | 1,000 | 1,000 | 1,000 |

## C.2. Training Details

The details of generation pipelines and classifier training are shown in Tab. 16 and Tab. 17, respectively.

*Table 16.* Details of generation pipelines.

| config | value |
|---|---|
| model | stable diffusion 2.0 model (Rombach et al., 2022) |
| prompt $\mathcal{C}_y$ | A photo of a [CLASS] |
| inference steps | 20 |
| text guidance strength | 7.5 |
| random noise strength (positive pipeline) ($\sigma_t$) | 1.0 |
| random noise strength (negative pipeline) ($\sigma_t$) | 0.2 |

*Table 17.* Details of classifier training.

| config | value |
|---|---|
| model | WideResNet-28-2 (Zagoruyko & Komodakis, 2016) |
| data augmentation | random horizontal flipping and normalization |
| batch normalization | optimized over the initial 100 iterations |
| optimizer | Adam |
| epoch | 400 |
| input size | $32 \times 32$ |
| batch size | 32 |
| learning rate | $5 \times 10^{-3}$ |
| loss weight $\lambda_1$ | 1 |
| loss weight $\lambda_2$(CIFAR-10) | 2 |
| loss weight $\lambda_2$(CIFAR-100) | 5 |
| loss weight $\lambda_2$(Tiny-ImageNet) | 20 |
| interval for confidence-based labeling (in epochs) | every 40 epochs |
| confidence-based labeling round | 10 |

