# OpenReview forum: "Unsupervised Learning for Class Distribution Mismatch"
_ICML.cc/2025/Conference — ICML 2025 poster_

### Official Review · Reviewer_Zj2u · 2025-03-05

**Overall Recommendation:** 4

**Summary:**

This paper proposes an Unsupervised Learning for Class Distribution Mismatch (UCDM), which constructs positive-negative pairs from unlabeled data for classifier training. The method randomly samples images and uses a diffusion model to add or erase semantic classes, synthesizing diverse training pairs. Extensive experiments on three datasets demonstrate UCDM’s superiority over previous semi-supervised methods.

## update after rebuttal
I appreciate the authors' detailed response. I will maintain the current positive score.

**Claims And Evidence:**

Yes, as the authors have stated, the experimental results presented demonstrate UCDM’s superiority over previous semi-supervised methods. Specifically, with a 60% mismatch proportion on Tiny-ImageNet dataset, our approach, without relying on labeled data, surpasses OpenMatch (with 40 labels per class) by 35.1%, 63.7%, and 72.5% in classifying known, unknown, and new classes.

**Essential References Not Discussed:**

No

**Experimental Designs Or Analyses:**

Yes, it is a reasonable design to test separately on closed-set and open-set tasks.

**Methods And Evaluation Criteria:**

Yes, exploring class distribution mismatch in an unsupervised manner does have certain significance.

**Other Comments Or Suggestions:**

No

**Other Strengths And Weaknesses:**

The overall description of the manuscript is clear, and unsupervised class distribution mismatch is a meaningful task. For small Weaknesses, "Imagenet: A large-scale hierarchical image database." appears twice in the references.

**Questions For Authors:**

1. How can authors effectively ensure that the generated images are truly positive or negative samples for generating tasks?
2. Have the authors explored how to generate hard samples in the proposed method, namely those that are more helpful in improving model performance?

**Relation To Broader Scientific Literature:**

Class distribution mismatch is a valuable field, and solving this problem in an unsupervised manner is a direction worth exploring. However, due to my limited knowledge in this field, I am unable to identify the relationship between this work and other existing works.

**Theoretical Claims:**

Yes, the proposed method operates without ground truth labels in the training data and utilizes only a predefined set of class names from known classes.

---

> ### Author Rebuttal · Authors · 2025-03-31
>
> > Q1: For small Weaknesses, "Imagenet: A large-scale hierarchical image database." appears twice in the references.
>
> Thank you for your careful review. We will correct this in the final version.
>
>
> > Q2: How can authors effectively ensure that the generated images are truly positive or negative samples for generating tasks?
>
> Thank you for your insightful question.
>
> Our **theoretical analysis** ensures that the generated images are truly positive or negative samples. Furthermore, our **experimental results** as well as **visualizations** further verify this. The details are presented below:
>
> + **Theoretical analysis:** Theorem 3.1 and 3.2 guarentee that our method can erase semantics in images for negative instance generation. For positive instances, we exploit a conditional diffusion model, starting from a seed sample-based initialization, to ensure the generated images resemble data distributions and exhibit the target semantics.
> + **Experimental results:** If the generated images contained a high proportion of false positives or false negatives, the performance would be unsatisfactory. As shown in the ablation study in Figure 4(a), **utilizing only the generated images to train the classifier improves performance**, quantitatively confirming the correctness of the generated images.
> + **Visualizations:** Section B.13 in the Appendix demonstrates that the generated images **closely align with the expected characteristics** of positive and negative samples, providing qualitative validation of the label correctness of the generated images.
>
> **Conclusion:**
>
> In the revision, we will **include the above analysis in the conclusion of Section 3.4 as follows:** "The theoretical analysis outlined in Theorems 3.1 and 3.2, along with the diffusion-driven approach for positive instance generation, confirms the reliability of both the generated negative and positive instances. This is further supported by the experimental results in Sec. 4.3 and the visualizations presented in Appendix B.13.".
>
> > Q3: Have the authors explored how to generate hard samples in the proposed method, namely those that are more helpful in improving model performance?
>
> Thank you for your insightful question!
>
> We incorporate hard samples into training through a **confidence-based labeling module that identified difficult real instances**, as shown in Section 3.5. These hard examples are then paired with generated images for training. By progressively **lowering the confidence threshold, increasingly difficult examples** are introduced during training. The results of our exploration of this approach are presented below:
>
> **Experimental results:**
>
> + The ablation study results (Figure 4(a)) show that integrating these **hard training pairs effectively improves performance** (pink bar), particularly when compared to training solely with generated images (green bar).
> + As shown in Figure 4(c), incorporating **an excessive number of harder samples** (by lowering the confidence threshold) **leads to unstable performance**. This is because the pseudo-labels for hard samples are assigned based on the classifier's predictions. When too many hard samples are included, the risk of incorrect pseudo-labels increases, leading to performance instability.
>
> **Future exploration:**
>
> Thank you for the valuable suggestion. Your insight motivates us to further refine **positive and negative instance generation** to construct **harder training pairs**. This can be achieved by **designing more specific prompts** to erase critical features, improving the contrast between positive and negative samples.
>
> We **have already generated highly similar positive and negative pairs** and provide examples in https://anonymous.4open.science/r/Rebuttal-UCDM-7787/Figure-Generated%20hard%20training%20pairs.pdf. We plan to further explore this direction in future work.

---

> > ### Comment · Reviewer_Zj2u · 2025-04-04
> >
> > I appreciate the authors' detailed response. I will maintain the current positive score.

---

> > > ### Author Response · Authors · 2025-04-04
> > >
> > > We sincerely appreciate your thoughtful review and will carefully incorporate all your suggestions in the revised version.
> > >
> > > Please feel free to let us know if you have any further questions—we'd be happy to address them.

---

### Official Review · Reviewer_d4vy · 2025-03-12

**Overall Recommendation:** 3

**Summary:**

The paper deals with learning with synthetic data from a stable diffusion model with labels that are obtained from prompting the model. The generated labels are only partially available to the model, i.e. the data is split in three subsets "known", "unknown" and "knew", where the known label categories are available during training time, while unknown and new labels are not known. However, data from the unknown category is available during training, while that of 'new' aren't. The main contribution of the authors approach is the generation of positive and negative pairs, on which the classifier then is trained contrastively. These pairs are obtained by a novel noising / denoising strategy, where for the negative sample the denoising is performed with respect to a non task specific score function in the diffusion process, whereas for the positive sampling the denoising is conditional. The formulae for the respective drift terms in the stochastic equations are derived.
The authors then eveluate their models on three datasets (Cifar10, Cifar100 and Tiny imageNet). The task is to classify the knowns correctly and to push the unknowns and new categories in an unknown class. The authors compare their model with various competitor models and find competitive performance for their own model that is often as good or better than the SOTA. Ablation studies on the various components are provided.

# # will raise score by one, as the authors provided new evidence that the noise terms are nearly constant in practice supporting their assumptions. Also the mathematical notation in the proofs has improved, although I'm still lacking a formal argument, why the delta-terms should be small  (and in which sense)

**Claims And Evidence:**

The claims are supported by mathematical theorems, which are however partially require assumptions on the noise estimators that are not checked and are rather loosely formulated (noise estimators over several steps are approximately constant). The numerical evidence given is a little strange, as many competitor models do not work at all on some sub-tasks (0-precision entries in tables), which permits the question if the comparison with these models is well chosen.

**Essential References Not Discussed:**

None that I have in mind.

**Experimental Designs Or Analyses:**

The Experiments only deal with rather small image sizes - experiments on the proper ImageNet would have made the case stronger.

**Methods And Evaluation Criteria:**

The evaluation on three datasets is ok, but all of them are small. Some competitor models seem not to perform at all on the given task.

**Other Comments Or Suggestions:**

If the conditions of the theorems can not be properly stated, one perhaps should not call these theorems. An at least empirical checking of the validity of assumptions would be solicited. I have the feeling that a strorng DG metod could be rather competitive in this field.

The style of writing is sometimes unclear and should be improved.

This review should be seen as a low confidence review.

**Other Strengths And Weaknesses:**

The set up here is close to domain generalization from syn to real and a look into the literature in this feield would be a nice idea.

**Questions For Authors:**

None.

**Relation To Broader Scientific Literature:**

The relation to the broader literature is well done.

**Theoretical Claims:**

I found the Theorems 3.1 and 3.2 hard to evaluate, as the validity of the assumptions is unclear. Theorem 3.1 is from prior work. The calculation leading to Theorem 3.2 itself is ok.

---

> ### Author Rebuttal · Authors · 2025-03-31
>
> > Q1:Unvalidated assumptions of noise estimators.
>
> Thanks for your question.
>
> The assumption **follows existing works[1,2,3]**. We further **verify its validity** as follows.
> + **Same assumption in DDIM and DDIM inversion.** DDIM[1] solves diffusion ODEs via **forward Euler, where $\epsilon(\mathbf{x}\_t,t)\approx\epsilon(\mathbf{x}\_{t-1},t)$**. DDIM inversion applies the same in reverse[2,3]. The approximation quality depends on $\mathbf{x}\_t-\mathbf{x}\_{t-1}$ and $\epsilon\_{\theta}$’s sensitivity to $\mathbf{x}\_t$[2].
> + **Validation.** We analyze the **discrepancy between $\epsilon(\mathbf{x}\_t, t)$ and $\epsilon(\mathbf{x}\_{t-1}, t)$**, measured by ***1-cosine similarity***, over 20 DDIM steps. Results show **near-perfect alignment**, consistent with **Fig.S10 in [3]**.
> step|1|2|3|4|5|6|7|8|9|10|11|...|14|15|...|19|20
> -|-|-|-|-|-|-|-|-|-|-|-|-|-|-|-|-|-
> **condition**|4e-2|3e-2|2e-2|2e-2|1e-2|8e-3|5e-3|3e-3|3e-3|2e-3|2e-3|...|2e-3|1e-3|...|1e-3|4e-4
> **uncondition**|4e-2|3e-2|2e-2|2e-2|1e-2|8e-3|5e-3|3e-3|3e-3|3e-3|2e-3|...|2e-3|1e-3|...|1e-3|4e-4
>
> In the revision, we will **supplement this validation to Appendix A.7**.
>
> **Reference:**
>
> [1] Denoising Diffusion Implicit Models,ICLR 2021.
>
> [2] Fixed-point inversion for text-to-image diffusion models,CoRR 2023.
>
> [3] Edict: Exact diffusion inversion via coupled transformations,CVPR 2023.
>
> > Q2:Competitor models yield 0 precision on sub-tasks, raising model selection concerns.
>
> Thanks for your question.
> + **Our method and all compared ones focus on class distribution mismatch(CDM)**, ensuring a fair and reasonable comparison.
> + **Two key factors** explain the poor performance:
> method|reason for poor accuracy
> -|-
> $\text{DS}^3\text{L}$, UASD, CCSSL, T2T|**Designed for closed-set task**, unable to handle unknown or new classes
> MCTF, IOMatch, OpenMatch|**Dependence on labeled data** causes imbalanced accuracy across known, unknown, and new classes
> + The **Unsupervised CDM setting is newly proposed**, and the performance gap highlights our method's strength in tackling this problem.
>
> > Q3: Small size of used datasets.
>
> Thanks for your suggestion.
>
> While we **follow the same dataset evaluation as recent works**, we have additionally **evaluate on a larger dataset** to further validate the effectiveness of our method, as recommended.
> + **Dataset selection:** The datasets used are common in recent studies, ensuring fairness. ImageNet-30(39k images, 30 classes) used in some methods are smaller than Tiny-ImageNet(100k images, 200 classes) in our study.
> method|CIFAR10|CIFAR100|Tiny-ImageNet|Larger dataset
> -|-|-|-|-
> $\text{DS}^3\text{L}$|✅|❌|❌|❌
> IOMatch|✅|✅|ImageNet-30|❌
> OpenMatch|✅|✅|ImageNet-30|❌
> MCTF|✅|❌|✅|❌
> UASD|✅|✅|✅|❌
> ours|✅|✅|✅|✅(below)
> + **Large-scale evaluation:** We test on 763,577 images from a combination of CIFAR10, SVHN, Flower-102, and Food-101 datasets. Results show our method **performs well on large datasets**.
> method|close(acc.)|open(kno.)|open(unkno.)|open(new.)|open(bala.)
> -|-|-|-|-|-
> CCSSL|57.7|57.7|0|0|-14.1
> T2T|60|60|0|0|-15.6
> IOMatch|46.7|20.4|36.3|75.6|15.7
> OpenMatch|19.8|17.7|18.9|12.5|13
> ours|53|48.3|100|83.4|50.8
>
> > Q4:Small image sizes in experiments.
>
> Thanks for your suggestion.
>
> The image size **follows prior works** like MCTF and UASD. As recommended, we **test on Tiny-ImageNet with $224 \times 224 \times 3$**, and the results show **similar trends** to those in the paper:
> method|close(acc.)|open(kno.)|open(unkno.)|open(new.)|open(bala.)
> -|-|-|-|-|-
> CCSSL|24.2|24.2|0|0|-5.9
> T2T|27.5|0|0|0|-6.7
> IOMatch|31.8|5.8|96.6|96.6|13.9
> OpenMatch|9.5|8.6|6.4|7.8|6.5
> ours|35|14.5|88.8|87.8|21.1
>
> > Q5:Relation to domain generalization(DG) and its competitiveness.
>
> Thanks for your suggestion.
>
> We compare our setup with DG and open DG to clarify key differences and also compare it with a representative open DG method.
>
> + **Relation to DG.**  DG handles **domain shifts** (e.g., cartoon vs. natural images) **without class mismatch**. Open DG adds class mismatch but **requires labeled data from multiple domains**, while our setup is based entirely on unlabeled data.
> problem|unlabeled data|training set domains|train-test class mismatch|data shift
> -|-|-|-|-
> DG|❌|≥ 1|❌|✅
> Open DG|❌|>1|✅|✅
> ours|✅|1|✅|❌
> + **Comparison with open DG method DAML[1]** on cross-dataset. Unlike ours, **DAML is fully supervised** with ground truth for all data, including unknown classes—an ideal but **unrealistic scenario for unsupervised methods**. Even so, our method achieves **comparable accuracy on unknown and new classes**, showing its effectiveness without labeled data.
> method|close(acc.)|open(kno.)|open(unkno.)|open(new.)|open(bala.)
> -|-|-|-|-|-
> DAML|81.5|62.3|99.9|85.5|63.6
> ours|53|48.3|100|83.4|50.8
>
> We appreciate this suggestion and will **supplement a DG review and this experiment in the revised version**.
>
> **Reference:**
>
> [1] Open domain generalization with domain-augmented meta-learning,CVPR 2021.

---

> > ### Comment · Reviewer_d4vy · 2025-04-05
> >
> > The assumptions on almost constant noise have been better explained, also the experiments on larger image sizes strengthen the paper. The way the theorems are 'proven' with the approx -sign which can be everything and nothing is now better justified but still not fully convincing to me. Nevertheless, the paper has improved and I would raise my score to borderline.

---

> > > ### Author Response · Authors · 2025-04-06
> > >
> > > Dear Reviewer d4vy,
> > >
> > > We sincerely appreciate your thoughtful feedback and consideration in raising your score. We are very glad that our previous responses helped clarify the assumptions and experimental results.
> > >
> > > We understand your concern regarding the use of the **approximation symbol** in our theoretical analysis. To address this, we **refine the assumption $\epsilon(\mathbf{x}\_t, t) \approx \epsilon(\mathbf{x}\_{t-1}, t)$ by explicitly expressing their difference as:**
> > > $$
> > > \epsilon(\mathbf{x}\_t, t) - \epsilon(\mathbf{x}\_{t-1}, t) = \delta\_t,
> > > $$
> > > where $\delta\_t \in \mathbb{R}^{64 \times 64}$ has the same shape as $\mathbf{x}\_t$ and characterizes the element-wise deviation between the two noise estimates.
> > >
> > > + Accordingly, **Equation (5) in Theorem 3.1 is rewritten as:**
> > > $$
> > > \mathbf{x}\_t = \sqrt{\alpha\_t} \mathbf{x}\_0 - \sum_{i=0}^{t-1} \left[ \nabla\_{\mathbf{x}\_i} \log p\_{\theta}(\mathbf{x}\_i)^{s\_i} + \nabla_{\mathbf{x}\_i} \log p\_{\theta}(y \mid \mathbf{x}\_i)^{s\_i} \right] + \sum\_{i=0}^{t-1} \frac{s\_i}{1 - \sqrt{\bar{\alpha}\_{i+1}}} \delta\_{i+1}.
> > > $$
> > > The **smaller the values in $\delta\_i$**, the **more accurately $\mathbf{x}\_t$** follows the idealized trajectory defined by the deterministic components above.
> > >
> > > + Similarly, **Equation (7) in Theorem 3.2 is rewritten as:**
> > > $$
> > > \tilde{\mathbf{x}}\_0 = \mathbf{x}\_0 - \frac{1}{\sqrt{\alpha\_t}} \sum\_{i=0}^{t-1} \nabla\_{\mathbf{x}\_i} \log p\_{\theta}(y|\mathbf{x}\_i)^{s\_i} + \sum\_{i=1}^{t-1} \sum\_{j=i}^{t-1} \frac{s\_i}{\sqrt{\alpha\_t(1 - \bar\alpha\_{j+1})}} \left[ \tilde\delta\_{j+1} - \delta\_{j+1} \right],
> > > $$
> > > where $\tilde\delta\_i = \epsilon\_{\theta}(\tilde{\mathbf{x}}\_i, i) - \epsilon\_{\theta}(\tilde{\mathbf{x}}\_{i-1}, i)$. The **smaller the magnitude of $|\tilde\delta\_{j+1} - \delta\_{j+1}|$**, the **better the reconstruction of $\mathbf{x}\_0$**, and the **more faithfully** the visual characteristics are preserved.
> > >
> > > + The **full derivation** is made available at https://anonymous.4open.science/r/Rebuttal-UCDM-7787/Proof.pdf; please feel free to check it at your convenience.
> > >
> > > **Analysis:**
> > > - **This refinement does not affect the rest of the analysis in the paper**, as it just makes the approximation $\epsilon(\mathbf{x}\_t, t) \approx \epsilon(\mathbf{x}\_{t-1}, t)$ explicit. However, it **improves the mathematical rigor** and enables us to quantify the potential impact of $\delta\_i$ in a more principled way.
> > >
> > > - **In our experiments, we set $\delta_i = 0$ under the forward Euler update**, which is supported by the empirical observation that $\epsilon(\mathbf{x}\_t, t)$ and $\epsilon(\mathbf{x}\_{t-1}, t)$ are nearly identical in practice.
> > >
> > > **In the revision, we will update Theorem 3.1 and Theorem 3.3, along with their corresponding proofs**, as shown above, to improve the clarity and rigor of our analysis.
> > >
> > > If there are any remaining concerns regarding the theorems or other parts of the paper, we would be more than happy to address them in further revisions.
> > >
> > > Thank you again for your constructive suggestions and support!

---

### Official Review · Reviewer_ikXz · 2025-03-13

**Overall Recommendation:** 2

**Summary:**

The paper addresses the problem of class distribution mismatch (CDM), where training and target task class distributions differ.
Previous methods rely on labeled data in semi-supervised settings, limiting applicability.
The authors propose ​Unsupervised Learning for CDM (UCDM), which uses a diffusion model to synthesize positive-negative instance pairs from unlabeled data.

**Claims And Evidence:**

Yes, the claims are supported by clear and convincing evidence.

**Superiority over semi-supervised methods**: Results in Tables 1–4 show consistent improvements across datasets and mismatch proportions.

**Label-free effectiveness**: Ablation in Fig. 4(b) demonstrates UCDM outperforms labeled baselines.

**Essential References Not Discussed:**

No critical omissions detected in cited literature.

**Experimental Designs Or Analyses:**

Yes, the experimental designs and analyses are sound and valid.

**Comprehensive evaluation**: Covers both closed/open-set tasks across multiple datasets.

**Ablation studies**: Tests loss components (Fig. 4a) and instance generation strategies (Table 5).

**Methods And Evaluation Criteria:**

My primary concern lies in the **rationale behind the proposed motivation**: While positioned as an unsupervised method, the use of a conditional diffusion model to generate positive-negative instance pairs appears to implicitly incorporate label information through the class-conditional generation process.
This creates potential ambiguity in maintaining true unsupervised learning principles, as conventional unsupervised approaches (e.g., self-supervised learning) typically evaluate through linear probing without explicit class guidance during representation learning.

**Other Comments Or Suggestions:**

No additional comments.

**Other Strengths And Weaknesses:**

Figure clarity: Figure 3 is overly complex and difficult to follow; simplifying the training diagram would improve readability.

**Questions For Authors:**

None.

**Relation To Broader Scientific Literature:**

The work connects to: Semi-supervised CDM methods and Diffusion models.

**Theoretical Claims:**

Yes, I checked the correctness of the proof for the Theorem 3.1.

---

> ### Author Rebuttal · Authors · 2025-03-31
>
> > Q1: Rationale behind the proposed motivation: While positioned as an unsupervised method, the use of a conditional diffusion model to generate positive-negative instance pairs appears to implicitly incorporate label information through the class-conditional generation process. This creates potential ambiguity in maintaining true unsupervised learning principles, as conventional unsupervised approaches (e.g., self-supervised learning) typically evaluate through linear probing without explicit class guidance during representation learning.
>
> Thank you for your insightful question.
>
> Our problem and method align with unsupervised learning **as defined in Deep Learning[1]**. Additionally, **class-conditional generation** and **explicit class guidance** in unsupervised learning have been extensively **explored in existing studies** across various tasks.
>
> + **Definition of unsupervised learning**: Our method trains the classifier with generated images and pseudo-labels **without any human annotation**, **following Deep Learning[1]**, which defines unsupervised learning as "most attempts to extract information from a distribution that does not require human labor to annotate examples."
>
> + **With class-conditional generation:** Our approach aligns with **unsupervised domain adaptation[4,5]**, where **class names** and **conditional diffusion models generate target data** for training. This further confirms that introducing class-conditional generation is still considered unsupervised.
>
> + **With explicit class guidance in unsupervised learning:** The "class name" setting in our method is commonly adopted in the unsupervised fine-tuning of multimodal models [2,3], where models are adapted to unlabeled target data, assuming **target class names are known**, but the mapping to the unlabeled data is not. Accordingly, this assumption is rational as well as widespread.
>
> We summarize the **commonalities** between existing unsupervised studies and our work as follows.
>
> method  | class names known | unsupervised |conditional diffusion model
> -|-|-|-
> Pouf [2]| ✅ | ✅ | ❌（pretrained CLIP）
> UEO [3]|✅ | ✅ | ❌（pretrained CLIP）
> DATUM [4] | ✅ | ✅ | ✅
> DACDM [5] | ✅ | ✅ | ✅
> ours | ✅ | ✅ | ✅
>
>
> **Conclusion:**
>
> Our method adheres to the principles of unsupervised learning, as it **does not rely on manually annotated labels**.
>
>
> We greatly appreciate your valuable feedback, we will **supplement the following description in paragraph 4 of Section 1:** "In this context, we aim to construct positive-negative pairs for training the classifier without any human annotation, adhering to the unsupervised learning setting [1].".
>
>
>
> **Reference:**
>
> [1] Deep learning. Cambridge: MIT press 2016.
>
> [2] Pouf: Prompt-oriented unsupervised fine-tuning for large pre-trained models. ICML 2023.
>
> [3] Realistic Unsupervised CLIP Fine-tuning with Universal Entropy Optimization. ICML 2024.
>
> [4] One-shot unsupervised domain adaptation with personalized diffusion models. CVPR 2023.
>
> [5] Domain-guided conditional diffusion model for unsupervised domain adaptation.
> Neural Networks 2025.
>
>
> > Q2: Figure clarity: Figure 3 is overly complex and difficult to follow; simplifying the training diagram would improve readability.
>
>
> Thank you for your helpful suggestion.
>
> Figure 3 illustrates the **classifier training pipeline** based on unlabeled data and generated instances.  To facilitate readability, we have **simplified the figure in the revised version, highlighting the following three stages**:
>    - **Stage 1:** Generated positive and negative instances are used to **create training pairs** for classifier training, following Eq. (9).
>    - **Stage 2:** The trained classifier(frozen) **selects confident real images from unlabeled data for pseudo-labeling**, using Eq. (10) and Eq. (11).
>    - **Stage 3:** Training pairs are **constructed using both selected and generated data**, and the classifier is further trained following Eq. (12), similar to Stage 1.
>
> The updated figure can be found at https://anonymous.4open.science/r/Rebuttal-UCDM-7787/Figure3-Simplified%20framework.pdf. Please feel free to check it.

---

### Decision · Program_Chairs · 2025-05-01

**Decision:**

Accept (poster)

**Comment:**

The paper uses a diffusion model to construct positive-negative pairs from unlabeled data for classifier training to handle class distribution mismatch. The paper received three reviews and an author rebuttal was submitted. The basis of noise estimator approximations in the theoretical justifications is initially questioned by d4vy, but deemed to be satisfactory with the rebuttal clarifications. The additional experimental results for larger datasets and higher resolution images were also acknowledged to be sufficient. Zj2u supports acceptance based on the presentation and the author rebuttal well-addresses the question on utilization of hard samples. The validity of calling the method unsupervised is questioned by ikXz given the use of a conditional diffusion model that may incorporate the label definition, which the author rebuttal clarifies as a terminology used in prior literature with the intent that no human annotation is used. This is an acceptable position, but the authors are encouraged to include the rebuttal clarifications in a revised version. In balance, the ACs agree with the reviewer majority that the paper brings novel technical contributions and is sufficiently supported by empirical observations. So, the paper is recommended for acceptance at ICML. The authors are encouraged to include the further clarifications and experiments from the rebuttal in the final version of the paper.